



# Impacts of spatial heterogeneity of anthropogenic aerosol emissions in a regionally-refined global aerosol-climate model

Taufiq Hassan[1], Kai Zhang[1], Jianfeng Li[1], Balwinder Singh[1], Shixuan Zhang[1], Hailong Wang[1], and Po-Lun Ma[1]

[1]Atmospheric Sciences and Global Change Division, Pacific Northwest National Laboratory, Richland, WA, USA

**Correspondence:** Taufiq Hassan (taufiq.hassan@pnnl.gov) and Kai Zhang (kai.zhang@pnnl.gov)

**Abstract.** Emissions of anthropogenic aerosol and their precursors are often prescribed in global aerosol models. Most of these emissions are spatially heterogeneous at model grid scales. When remapped from low-resolution data, the spatial heterogeneity in emissions can be lost, leading to large errors in the simulation. It can also cause the conservation problem if non-conservative remapping is used. The default anthropogenic emission treatment in Energy Exascale Earth System Model (E3SM) is subject to both problems. In this study, we introduce a revised emission treatment for the E3SM atmosphere model (EAM) that ensures conservation of mass fluxes and preserves the original emission heterogeneity at the model-resolved grid scale. We assess the error estimates associated with the default emission treatment and the impact of improved heterogeneity and mass conservation in both globally uniform standard-resolution (∼165 km) and regionally-refined high-resolution (∼42 km) simulations. The default treatment incurs significant errors near surface, particularly over sharp emission gradient zones. Much larger errors are observed in high-resolution simulations. It substantially underestimates the aerosol burden, surface concentration, and aerosol sources over highly polluted regions, while overestimates these quantities over less-polluted adjacent areas. Large errors can persist at higher elevation for daily mean estimates, which can affect aerosol extinction profiles and aerosol optical depth (AOD). We find that the revised treatment significantly improves the accuracy of the aerosol emissions from surface and elevated sources near sharp spatial gradient regions, with significant improvement in the spatial heterogeneity and variability of simulated surface concentration in high-resolution simulations. In the next-generation E3SM running at convection-permitting scales where the resolved spatial heterogeneity is significantly increased, the revised emission treatment is expected to be better represent the aerosol emissions as well as their lifecycle and impacts on climate.

## 1 Introduction

The presence of lower tropospheric aerosols has significant impacts on air quality. Additionally, aerosols play a crucial role in the energy balance of the Earth system, as they can scatter or absorb radiation and affect the formation, lifetime, and albedo of clouds. Anthropogenic activities, including air pollution, have contributed to a substantial increase in the tropospheric aerosol burden since the pre-industrial era, further intensifying these effects (Bond et al., 2007). According to the report on the 6th Intergovernmental Panel on Climate Change (IPCC), the effective radiative forcing of anthropogenic aerosols ranges from -0.63 to -1.37 W m-2 (Smith et al., 2020). Furthermore, recent urban-scale studies have shown that anthropogenic aerosols



may impact regional climate by affecting the urban heat island intensity (Han et al., 2020; Yang et al., 2020; Wu et al., 2017) and altering the precipitating systems over urban areas (Rosenfeld et al., 2008; Van Den Heever and Cotton, 2007). Despite the crucial role of anthropogenic aerosols in affecting regional and global climate, significant uncertainties persist in their numerical simulation. One of the large uncertainties is how anthropogenic aerosol emissions are treated in the model (Textor et al., 2006).

Accurate representation of anthropogenic aerosol emissions and their time evolution are crucial for Earth system models (ESMs) and atmospheric chemistry and transport models (Hoesly et al., 2018). Most of these emission sources are spatially heterogeneous at regional and local scales, and their relatively short lifetime in the atmosphere results in a heterogeneous distribution, both geographically and vertically (Koch et al., 2009). For high−resolution simulations, representing this spatial heterogeneity, particularly near emission hot-spots, is important to improve aerosol simulation accuracy (Wang et al.,

2018, 2014). Furthermore, retaining this heterogeneity can be important since evaluating the model fidelity in areas such as aerosol formation, transport, cloud interactions, and deposition may depend on the accuracy of the prescribed emissions used to drive the models. Using lower-resolution data may result in a loss of heterogeneity, leading to lower accuracy in the aerosol simulation and model evaluations, particularly near the sharp emission gradient zones. This error may be more significant for high-resolution model simulations, often used in regional and urban-scale studies.

The U.S. Department of Energy (DOE) Energy Exascale Earth System Model (E3SM) is a state-of-the-art ESM that aims to produce actionable and accurate predictions of regional trends relevant to Earth system variability and change (Golaz et al., 2019, 2022). The E3SM atmosphere model (EAM, Rasch et al., 2019) has a rather comprehensive representation of physical and chemical aerosol processes, which simulates the lifecycle of aerosols and their interactions with clouds and radiation (Wang et al., 2020). However, for the emission of anthropogenic aerosols and their precursors, the default treatment used in

the standard and high-resolution EAM configurations has some limitations. First, the current anthropogenic emission treatment does not conserve mass. EAM uses an unstructured cubed-sphere spectral-element (SE) grid due to its advantages over regular latitude-longitude (RLL) grids, such as offering high-resolution capabilities and improved computational scalability through Regionally Refined Model (RRM) (Taylor and Fournier, 2010; Dennis et al., 2012). Since online conservative remapping of external forcing data (including emissions) is not currently available in E3SM, EAM linearly interpolates the data to the

model-native SE grid, and conservation is lost during the remapping. Second, with the current emission treatment, it is hard to preserve the spatial heterogeneity as in the original emission data. EAM uses low-resolution ($\sim2°$) anthropogenic aerosol emission data for simulations performed at the standard resolution or simulations with RRM, so the linear interpolation will cause large errors (see section 3.1 for details). On the other hand, linearly interpolating high-resolution data to low resolution might cause large conservation errors. Therefore, improving the emission treatment in EAM is essential, especially for future

E3SM applications with regional refinement at high resolutions, such as convection-permitting scales.

Similar issues in other atmosphere models with unstructured grids are discussed in recent studies (Pfister et al., 2020; Schwantes et al., 2022) and the model-native emission data have been used for better simulating ozone and evaluating the sensitivity to horizontal resolutions. However, these studies did not investigate/report the model sensitivity to the emission change only (original vs. native-grid emissions) and evaluate the impact on simulated aerosol processes due to improved





representation of emission heterogeneity. In this study, we revise the anthropogenic emission treatment in E3SM and directly read/apply the conservatively-remapped emissions at the model's native (unstructured) grid. The improved treatment conserves the mass fluxes and it can also help preserve the emission heterogeneity when a high-resolution model or RRM is used. We assess the impact on the simulated aerosol mass concentrations, optical properties, and anthropogenic aerosol forcing estimate, with a focus on the impacts of improved heterogeneity and mass conservation in high-resolution simulations.

This article is organized as follows. In Section 2, we describe the original and revised emission implementations and how the emission data are prepared. We also provide an overview of the model simulations and configurations for this study. In Section 3, we estimate the error in model emission input, compare model simulations with different emission treatments, and evaluate them against available observations. Finally, we summarize our findings and provide conclusions in Section 4.

## 2 Methods

In this section, we first briefly describe the model used for this study. We then provide an introduction of the original emission implementation and describe how we improve it. The simulation design and observational data used to evaluate the model are also described in this section.

### 2.1 Model Overview

In this study, we utilize the atmosphere component (EAMv2) of the Energy Exascale Earth System Model version 2 (E3SMv2)
(Golaz et al., 2022). E3SMv2 has a comprehensive aerosol model representation (Wang et al., 2020), which is based on the four-mode version of Modal Aerosol Module (MAM4, Liu et al., 2016). It represents major anthropogenic and natural aerosol species with four lognormal size modes, including black carbon (BC), primary organic matter (POM), secondary organic aerosol (SOA), marine organic aerosol (MOA), sulfate ($SO_4$), mineral dust, and sea-salt (Wang et al., 2020). The model accounts for various aerosol processes including aerosol microphysics (condensation, nucleation, coagulation, aging, and aerosol
wateruptake), emissions, sulfur chemistry, simplified SOA formation, dry removal (dry deposition and sedimentation), and wet removal. The model also considers aerosol effects on radiation and cloud formation (Wang et al., 2020; Zhang et al., 2022a).

EAMv2 employs a spectral element dynamical core (Taylor and Fournier, 2010; Dennis et al., 2012) as in EAMv1 (Rasch et al., 2019), but utilizes "physics grid" (pg2 grid) for unresolved physics parameterization and a separate dynamics grid for resolved processes (Hannah et al., 2021). This allows for an increased computational efficiency by reducing the effective reso-
lution for the physics parameterization computations. The standard configuration of the model uses a low "ne30pg2" resolution (LR), which has a grid spacing of ∼110 km (∼1°) for dynamics and a grid spacing of ∼165 km (∼1.5°) for physics. E3SMv2 also supports fully coupled regionally refined mesh (RRM) configurations for high-resolution applications. The regionally refined configurations are computationally inexpensive than the uniform high-resolution setup and can be useful in identifying highly heterogeneous fields (Wu et al., 2018; Rahimi et al., 2019). One of the supported stable RRM configurations has a high-
resolution mesh centered over North America (NA RRM, Tang et al., 2022). The NA RRM setup has a horizontal resolution of ne120pg2 (∼28-km dynamics grid and ∼42-km physics grid) over North America and a horizontal resolution of ne30pg2



over rest of the globe (Fig. 1b). The high-resolution refined mesh is located approximately within 10°N to 80°N and 170°W to 10°W. This region covers a significant number of observation sites (AERONET and IMPROVE) for model evaluation (Fig. 1b). NA RRM has a total of 57816 computational elements or grid cells as opposed to 21600 in the standard configuration.

In the present study, we conduct experiments in both LR and NA RRM setups to explore the impact of the new emission treatment. More detailed description of the experimental setups is available in section 2.4.

## 2.2    The original aerosol emission treatment in E3SM

The E3SM accounts for emissions of both natural and anthropogenic aerosols, as well as their precursors (Wang et al., 2020). The emissions of mineral dust, sea salt, and MOA, which are wind-driven primary natural aerosols, are parameterized based 100    on several variables such as surface wind speed, soil erodibility, sea spray fluxes, and sea surface temperature (Zender et al., 2003; Burrows et al., 2022). In contrast, the emissions of BC, POM, $SO_4$, and precursor gases, are prescribed and read in from input data. Note that the emission of Dimethyl sulfate (DMS) is also prescribed, rather than interactively calculated based on surface wind speed and seawater DMS concentrations in some other global aerosol models (e.g. Zhang et al., 2012). The aerosol particles in these prescribed emission data can be distributed in Aitken mode, accumulation mode, or primary carbon 105    mode (Liu et al., 2012, 2016). All prescribed emissions of BC and POM are considered as primary carbon mode aerosol particles (Liu et al., 2016), while sulfate ($SO_4$) aerosol particles are prescribed as either Aitken or accumulation mode based on the emission sectors or types. Aerosol number emission fluxes are prescribed based on the mass fluxes and the assumed emission particle size distributions (see supplemental materials of Liu et al., 2012). All prescribed aerosol species are emitted as interstitial aerosols at the surface and/or elevated locations.

The anthropogenic emissions from agricultural, industrial, energy, transportation, and domestic sectors for EAMv2 are mostly derived from Community Emissions Data System (CEDS) inventory (Hoesly et al., 2018). The biomass burning emission is derived from the fourth generation of the Global Fire Emissions Database (GFED4, Giglio et al., 2013; Van Marle et al., 2017). CEDS provides historical (1750−2014) inventory of anthropogenic GHGs, reactive gases, and aerosols for Coupled Model Intercomparison Project phase 6 (CMIP6, Eyring et al., 2016). It includes anthropogenic emissions for the primary 115    aerosol species, such as BC and OC, and sulfur dioxide ($SO_2$, a gas precursor for sulfate). These data are in regular latitude longitude (RLL) grid at 0.5°×0.5° resolution. GFED4 is also a part of the inputs for CMIP6, which provides historical (1750−2015) anthropogenic aerosol emission inventory of BC, OC, and $SO_2$ from biomass burning in RLL grid at 0.25°×0.25° resolution. In the standard configuration of EAMv2, low resolution (1.9°×2.5°) RLL gridded monthly historical (1850-2014) emission data are prescribed for ne30pg2 and RRM simulations. EAM also considers elevated emissions from biomass burning, 120    industrial, energy, and volcanic sources. These emissions are distributed across 13 different altitudes, ranging from the ground level to approximately 7 km above the surface. The distribution of emission within each layer is uniform, but the distribution varies between layers, depending on the source of the emission. Elevated sulfur emissions from energy and industrial sectors are emitted at altitudes between 100 and 300 meters above the surface, while biomass burning emissions are distributed across all 13 altitude ranges based on the recommendation from Dentener et al. (2006).



The emission treatment is a key component of the post-coupler processes (see Fig. 2a, excluding the dust emission that is calculated in the land model). The EAMv2 utilizes unified routines to process all prescribed anthropogenic aerosol and precursor gas emissions, which are provided as RLL data (monthly by default). They are subsequently linearly interpolated to the native model grid and temporal interpolation is also applied (left panel in Fig. 2b). The aerosol microphysics modules simulate the formation, growth, and removal of aerosols in the atmosphere, including processes of nucleation, coagulation,

condensation, and aging. The elevated emissions are applied to the gas-phase chemistry calculations as external forcing terms, while the surface fluxes are updated before dry deposition (Fig. 2a).

As mentioned earlier, linear interpolation is applied to convert the prescribed RLL grid data to model-native grid in EAMv2 (and earlier versions). Although it's convenient to linearly interpolate the same emission data to model grid at different spatial resolutions, the current treatment does not conserve mass. Also, when interpolated to higher resolutions, large emission errors will occur compared to the original emission data.


### 2.3   E3SM revised emission treatment

To address the limitations of the default emission treatment, we revised the emission implementation in EAMv2 as follows (Fig. 2b):

1. We have modified aerosol emission routines, which allows EAM to read prescribed anthropogenic aerosol emissions in
140         both regular latitude-longitude (RLL) and model-native (SE) grids.

2. To preserve spatial heterogeneity and acheive mass conservation, we conservatively remap the RLL high-resolution emission data to the model-native grid at selected resolutions. In addition to remapping, this also involves generating the grids, weights, and mapping files, as well as making the remapped emission data compatible with the revised emission treatment. To simplify and automate this procedure, we developed a Python wrapper package (ggen) that utilizes exist-
145         ing grid-generation and remapping tools (TempestRemap, Ullrich and Taylor, 2015; Ullrich et al., 2016, and ncremap, Zender; 2008). This package is applicable to both surface and elevated emissions at any given resolution.

3. The original online linear interpolation is switched off when the new treatment is used to read the conservatively remapped data directly.

The revised treatment has been implemented in both EAMv1 and EAMv2 for scientific evaluations.

In this study, we use the $0.63° \times 0.47°$ RLL grid emission data as the high-resolution emission input for anthropogenic aerosols and precursor gases. The data resolution is close to the ne120pg2 resolution ($\sim$42km for the physics grid), which is the resolution used for refined regions in the NA RRM simulation. These data were prepared for globally-uniform high-resolution E3SM applications (Caldwell et al., 2019) using the original CEDS and GFEDv4 data sets as mentioned above. When needed, we can use emission inventories at higher resolutions for simulations at even finer scales.





## 2.4 Simulations

Two groups of E3SMv2 simulations at different horizontal resolutions were conducted to estimate the impacts of the new emission treatment in EAMv2 (Table 1). Each group consists of control simulations using the default emission treatment and spectral-element (SE) treatment simulations utilizing the revised emission treatment. The simulations were performed for both the low-resolution standard configuration of E3SMv2 (ne30pg2 or LR) and high-resolution (NA RRM) setup (Fig. 1) with 72 vertical layers. For the control simulations, we follow the standard configuration and use the default low-resolution (1.9x2.5°) prescribed emissions of aerosols and precursor gases in RLL grid. For SE simulations, we prepared anthropogenic aerosols and precursor gas emissions on model-native grid from a high-resolution inventory. We conducted simulations with both present-day (PD, year 2014) anthropogenic aerosol emissions, and pre-industrial (PI, year 1850) anthropogenic aerosol emissions. All simulations include active atmosphere and land with prescribed monthly mean SST and sea ice from 2016.

All simulations were conducted using a meteorological nudging method (Sun et al., 2019), in which the horizontal wind components (u and v) were nudged to the European Centre for Medium-Range Weather Forecasts (ECMWF) ERA5 reanalysis (Hersbach et al., 2020) with a relaxation timescale of 6 hours. The nudging data were prepared for both LR and RRM grids, following method described in Zhang et al. (2022b). Nudging the horizontal winds in these simulations can well constrain the large scale circulation (Zhang et al., 2014), so that we can 1) assess the impact of model parameterization changes with shorter simulations; and 2) carry out a more accurate (co-located) evaluation against in-situ measurements than using the free running simulations.

These simulations were performed from October 1st, 2015, to December 31st, 2016. The first three months from the year 2015 were discarded as a model spin-up period, and the remaining 12 months were used for the analysis in this study. The choice of simulation year was based on readily available ERA5 hourly reanalysis data for nudging and active Atmospheric Radiation Measurement (ARM) sites over North America for future evaluations.

## 2.5 Observation data

To evaluate the model's ability to simulate regional to local distributions of aerosols with the updated BC, POM, and $SO_4$ aerosols, and optical properties such as Aerosol Optical Depth (AOD) are compared to observational data from regional networks such as the Interagency Monitoring of PROtected Visual Environment (IMPROVE) and Aerosol Robotic NETwork (AERONET) (Fig. 1a). Only present-day (PD) simulations (LR-PD and RRM-PD) were used for the evaluations.

IMPROVE, is a network of aerosol monitoring stations located in protected areas in the United States, such as national parks and wilderness areas (Malm et al., 2004). This network measures aerosol properties such as surface concentrations, size distribution, composition, and optical properties. IMPROVE surface concentration measurements are only available over the United States and provide daily data 3 times a week. For this evaluation, we consider daily average surface concentration measurements from all available IMPROVE sites for each aerosol species in the year 2016. Measurements are available from over 150 sites for BC, organic carbon (OC), and sulfate aerosol surface concentration. We also multiply the observed OC by 1.4 before comparing against simulated POM. Since the IMPROVE measurements are for fine aerosol particles, we do





not consider simulated aerosols in coarse mode. We also applied a conversion factor of 96/115 (~0.83) to the simulated
sulfate concentration (MAM4 in E3SM assumes the sulfate composition is ammonium bisulfate) before comparing against the
IMPROVE measurements.

AERONET, on the other hand, a global network of ground-based sunphotometers, measures aerosol properties such as size
distribution, composition, and optical properties (Holben et al., 1998). For this evaluation, we used AOD spectral radiometer
daily mean measurements from over 120 active sites during the simulation year (e.g., 2016), which fall within the North
America high-resolution mesh (bounded by 15°N to 75°N and 55°W to 170°W) in the RRM setup.

The simulated daily mean data were spatiotemporally co-located with the observational data according to the time and
location of each active observational site. These daily average data were conditionally sampled based on EAMv2 emission
differences found between the default and revised treatment (see details in section 3.3). We consider the same sites for RRM and
LR simulation evaluations. Finally, monthly means of these spatiotemporally collocated data were used to compare simulated
and observed surface concentrations and AOD.

## 3   Results and discussions

### 3.1   Improving emissions in EAM

One primary goal of our revised emission treatment is to improve the accuracy of the emissions data utilized in the standard
LR and RRM simulations. The default EAM emissions treatment fails to preserve spatial heterogeneity or conserve mass,
resulting in substantial errors (as described in section 2.2). Figure 3 shows the spatial distribution of surface BC (a-c) and
column-integrated $SO_2$ emissions (d-f) over the Eastern and Western US for the year 2014 in the original high-resolution data,
the default emission treatment on the RRM grid, and the revised treatment on the RRM grid. The anthropogenic emissions of
BC and POM are dominated by surface sources. $SO_2$ is the primary precursor of sulfate aerosols and the emission is dominated
by the elevated sources over land. Figure 3 indicates how well the heterogeneity of prescribed surface and elevated emissions
are represented in the standard RRM simulations when compared against the original high-resolution data.

The original high-resolution data (Figure 3a, d) show highly heterogeneous emissions over land, largely driven by industrial,
energy, and transportation sectors. As expected, the default emission treatment fails to capture most of the heterogeneity over
sharp emission gradient zones (Fig. 3b, e). For example, panels b and e depict seven major cities (e.g., Boston, New York,
Chicago, Toronto, Montreal, Los Angeles, and San Francisco) with large anthropogenic BC and $SO_2$ emissions respectively.
The default treatment not only significantly underestimates emissions (~80%) from those cities, but also grossly misrepresents
emissions in the nearby regions. This can severely affect the accuracy of high-resolution studies in urban regions. Additionally,
the default treatment provides an inaccurate representation of emissions near the coasts, where regions of BC sources from
shipping sectors appear to come from comparatively large emissions from transport and industrial sectors over land. In general,
both surface and vertically distributed emissions in the default treatment fail to maintain emission land-sea contrast near the
coastlines. In contrast, panels c and f illustrate the spatial emissions distributions in the improved treatment, which accurately
preserves the original spatial heterogeneity both inland and near the coast. Supplementary Fig. S1 depicts the large error




regions in terms of the emission difference between the improved and default treatment from surface and elevated sources. It also indicates that larger errors, from sharper spatial gradients, exist near regions with larger emissions. For instance, the difference between surface BC and POM is much larger over the Eastern US, which is consistent with the larger surface emissions over the Eastern US. This is important to separate the regions with sharper spatial gradients in the later sections. The improved implementation, which retains both heterogeneity and mass conservation, provides a more accurate representation of surface and vertically distributed emissions in EAMv2. It also accurately applies prescribed emissions in major cities, making the revised treatment suitable for urban-scale studies when running at high resolutions.

To evaluate the loss of accuracy of the EAMv2 emissions in the default treatment, we calculate error estimates against the more accurate emissions data that are conservatively remapped from the original high-resolution data. Table 2 provides a summary of the error estimates for BC, POM, and $SO_4$ aerosol emissions data used in the EAMv2 RRM simulations, including area-weighted spatial mean (Mean), normalized mean bias (NMB), standard deviation (StdDev), normalized standard deviation (NStdDevB), root mean square error (RMSE), and normalized RMSE (N_RMSE) for the present-day (year 2014). The estimations for elevated sources are based on column-integrated values. Table S1 presents the error statistics for emissions in the LR simulations. The RRM configuration used in this study has a high-resolution mesh over North America (NA), with the largest sources of emissions occurring over the land and only emissions from the shipping sector occurring over the ocean. Therefore, we consider NA land surfaces bounded by 15N to 75N and 55W to 170W for the error estimates.

The default treatment for anthropogenic aerosol emissions in both LR and RRM simulations consistently yields large RMSE for all metrics. It is worth noting that the NMB, which indicates mass conservation errors, is generally small ($< 1\%$) for the global mean estimate. However, it can be considerably larger for regional estimates, such as BC over the northeast United States, where it can reach up to 25% (not shown). Over NA land areas, it varies from ~1 to ~10% for RRM and ~0.3 to ~2.5% for LR emissions. Since NMB is influenced by the magnitude of emissions, it tends to be larger for anthropogenic emission sources than for biomass burning sources. As a result, we observe larger NMB for surface emissions of BC and POM. On the other hand, it is larger for elevated sources of $SO_4$ emissions, which includes vertically distributed anthropogenic sources from industry and energy sectors as well as biomass burning sources. This is consistent with larger sulfate emission differences from elevated sources between the improved and default treatment (Fig. S1).

Regardless of the NMB values, we find that both surface and elevated sources exhibit large RMSE values. To compare the RMSE values across different species and sources, we use normalized RMSE (N_RMSE) values. We found consistently larger N_RMSE values (ranging from 54% to 84%) for EAMv2 RRM emissions compared to the LR emissions (ranging from 34% to 57%), with the largest N_RMSE for the elevated sources of sulfate aerosols.

Overall, the default treatment for anthropogenic aerosol emissions lead to large errors in the input emisssion in both LR and RRM simulations. Improved emissions data prepared for our revised emission treatment can resolve these issues by maintaining the spatial heterogeneity (at corresponding resolutions) and mass conservation.





## 3.2  Model−to−model comparison

In this section, we compare the simulated fields between SE-PD and PD simulations to evaluate the error estimates from the
default emission treatment and the impact of implementing the revised emission treatment. We constrain these comparisons
within regional to local scales over North America to encapsulate large differences within the high-resolution RRM mesh.

### 3.2.1  Simulated climatological means

In Fig. 4, we present the spatial distribution of surface concentration resulting from RRM-PD simulation using the default
emission treatment, along with the relative differences between RRM-SE-PD and RRM-PD. The results show significant
differences over North America with a normalized root mean squared error (N_RMSE) of 39%, 34%, and 12% for BC, POM,
and sulfate aerosols, respectively. While the absolute relative differences of BC and POM surface concentrations can reach up
to 50% in some regions, sulfate aerosols exhibit weaker relative differences ranging from 2-10%. This can be attributed to the
fact that prescribed sulfur emissions are mostly emitted from elevated sources, such as industrial and energy sectors, as opposed
to BC and POM emissions, which are primarily emitted at surface level. Furthermore, significant differences were found in
simulated surface concentrations for LR simulations, with normalized RMSE of 19%, 15%, and 8% for BC, POM, and sulfate
aerosols, respectively (Fig. S2). The weaker N_RMSE in LR simulations compared to RRM simulations is consistent with the
weaker N_RMSE found from the default surface emissions used in the LR experiments (Table S1).

Simulated surface concentration differences exhibit positive and negative bias regions, indicating patterns of sharp spatial
gradients. Although these patterns appear randomly distributed across North America, they are closely linked to the differences
between prescribed emissions from the new and default treatment (Fig. S1). To confirm this, we separated North America
into three distinct regions based on the prescribed emission differences between the RRM-SE-PD and RRM-PD simulations.
Masking was applied to distinguish regions with strong and weak errors from default emissions. Regions with aerosol emission
differences above the 75th percentile, below the 25th percentile, and within the 25th/75th percentiles over North America were
selected.

Figure 5 shows the relative differences between different simulated fields, including aerosol burden, surface concentration,
net aerosol sources, and sinks. Each field is masked based on their respective aerosol emission difference beyond and within
the 25th/75th percentiles. Sharp spatial gradients are predominantly found near highly polluted regions, and fields masked
by emission differences above (below) 75th (25th) percentiles reveal larger positive (negative) relative differences. While the
estimates vary among different aerosol species and fields, the relative difference can range from -90% to over 50%. In contrast,
simulated fields masked by emissions within 25th/75th percentiles, representing weaker emission gradients, show significantly
smaller relative differences, ranging from 0.3 to ∼5%. These results suggest that the seemingly random distribution of relative
differences in Fig. 4 is strongly linked to the emission differences between the new and default treatment. It also indicates
that errors in default emissions not only impact simulated surface concentrations but also aerosol burden (weak) and their
source-sinks (strong). A decomposed source-sink analysis is described in a later section.



Figure 5 also displays relative differences from major cities over North America with large anthropogenic aerosol emissions (as depicted in Fig. 3). These cities are located above the 75th percentile masked regions and display similar patterns, with significant positive or negative biases in the simulated aerosol burden, surface concentration, and aerosol source and sinks. This finding suggests that errors arising from default emissions can have a significant impact on the accuracy of high-resolution urban-scale simulations. We note that the simulated aerosol burden yields weaker relative differences compared to surface

concentration. This is expected since we are analyzing long-term annual means of column-integrated concentration (burden), which are influenced by several other processes, such as condensation-aging, coagulation, aqueous-phase cloud chemistry, depositions, vertical diffusion, and horizontal transport.

       Figure 6 illustrates the spatial distribution of simulated annual mean aerosol extinction and absorption at the model surface layer, as well as AOD and absorption AOD (AAOD). From long-term means, we found larger differences in simulated

surface aerosol concentration. Therefore, larger relative differences in annual mean aerosol extinction and absorption are constrained near the surface level with a normalized RMSE of 13% and 29% respectively. Our revised treatment improves the anthropogenic aerosol emissions, leading to larger differences in simulated absorption profiles near the surface compared to extinction profiles. Extinction profiles are influenced by natural aerosols such as sea-salt and dust, in addition to the anthropogenic aerosols. Notably, aerosol absorption profiles near the surface can reach a relative difference of approximately 50%

over major cities, southern Mexico, and northwest North America, which is consistent with the spatial distribution of emission differences between the new and default treatment.

       AOD and absorption AOD are column-integrated aerosol extinction and absorption, respectively, that are strongly influenced by processes such as aerosol chemistry, horizontal transport, and vertical diffusion. Our results indicate that the annual mean spatial distributions of these simulated fields do not show significant differences, with some exceptions over northwest North

America, which can be attributed to the unusually high biomass burning emissions of BC and POM during the 2014 Northwest Territories (NWT) fires. The summer of 2014 was the most severe fire season in NWT history, resulting in wildfires burning a record 3.4 million hectares and an estimated emission of $164 \pm 32$ Tg of carbon into the atmosphere (Veraverbeke et al., 2017; Kochtubajda et al., 2019). Biomass burning emissions are prescribed as elevated sources of anthropogenic aerosols in E3SM. Substantial errors from default emissions persist at higher elevations (Table 2). Since the present-day simulations are

conducted using the emissions from the year 2014, the simulated AAOD shows large relative differences over NWT, which may be driven by the persisting errors in elevated BC and POM emissions from default treatment.

### 3.2.2    Simulated high−frequency fields

While the column-integrated concentration (i.e., aerosol burden) may be less affected, larger discrepancies can arise at the surface level and at higher elevations in high-frequency concentration profiles. Figure 7 exhibits significant differences in

simulated daily mean BC concentration profiles and column integrated burden. Panels a-d (e-h) illustrate vertically distributed aerosol concentrations and column-integrated burden from highly (nearby less) polluted locations to demonstrate the simulated biases near sharp spatial emission gradient zones. The vertical profiles indicate that the larger differences can persist at higher elevations above the surface. Over polluted regions, the default emission treatment significantly underestimates surface and





elevated aerosol concentration within the boundary layer. On the other hand, over comparatively cleaner nearby regions, the
default emission treatment significantly overestimates surface and elevated aerosol concentrations within the boundary layer.
Due to the high variance in daily mean fields, as opposed to long-term monthly or annual means, we found significantly larger
errors in simulated high-frequency data. For instance, relative differences in simulated BC burden could reach up to 25-30%
on certain days (Fig. 7d).

High-frequency daily mean AOD and absorption AOD can have N_RMSE values of approximately 15-20% over North
America. Figure 8 shows the time series of high-frequency daily mean extinction and absorption profiles, as well as the time
evolution of AOD and AAOD during the month of July 2016. The distributions are shown over a highly polluted location as
in Fig. 7 (a-d). Consistent with the persistent aerosol concentration differences found at higher elevations (Fig. 7), significant
differences exist in the simulated extinction and absorption profiles within the boundary layer. Although the long-term annual
mean AOD and AAOD do not display significant errors over the highly polluted northeast US region, relative differences from
daily mean estimates could reach up to approximately 10-12% on certain days (Fig. 8 d, h). Our results highlight the potential
impact of errors from default emission treatment on high-frequency aerosol concentrations at higher elevations, which can
further influence aerosol extinction profiles and lead to significant errors in simulated AOD. This finding is particularly relevant
as short-term high-frequency fields are often used for urban-scale studies and model evaluations against observations.

### 3.2.3 Decomposed source-sink analysis

Results in section 3.2.1 suggest that using the default emission treatment can lead to significant errors in aerosol source and
sinks, despite having a weaker impact on column integrated aerosol burden (Fig. 5). In this section, we conduct an aerosol
source-sink analysis to estimate the potential errors propagated from default emissions into different processes during the
RRM simulations. Figure 9 presents stacked bar plots indicating the fractional distribution of major processes contributing to
the total simulated aerosol sources and sinks in the EAMv2 present-day RRM simulation. The analysis considers the North
American region (15°N to 75°N and 55°W to 170°W) and estimates the percent contributions from annual means of each
component that drives the simulated sources and sinks. The overall fractional distribution is very similar in LR simulations
(not shown).

The stacked bar plots in Fig. 9a reveal that prescribed emissions are the primary drivers of BC and POM sources, with a
surface (from anthropogenic sources) to elevated (from biomass burning sources) emission ratio of 7:3 for BC and 6:19 for
POM, respectively. In contrast, only about 5% of the total sulfate sources are driven by the prescribed sulfate emissions. Gas-
aerosol exchange and in-cloud aqueous-phase ($SO_4$) chemistry contribute to sulfate sources by ~22% and ~71%, respectively.
BC and POM sinks are evenly modulated by dry and wet depositions, with turbulent dry deposition accounting for most of the
dry deposition (~41% and ~36%), and in-cloud scavenging accounting for most of the wet deposition (~56% and ~60%).
Stratiform clouds modulate the larger portion of the in-cloud wet deposition. Conversely, sulfate removal has an uneven ~15%
and ~85% contributions from dry and wet depositions respectively, with the largest contribution from stratiform in-cloud
scavenging (~65%).





Figure 10 and Figure 11 present the error statistics (in terms of weighted N_RMSE) for the simulated source and sink terms in RRM-PD, taking RRM-SE-PD as the reference solution. The weights are determined by the fractional contributions of each process shown in Fig. 9, and the actual RMSE and N_RMSE for each process can be found in Table S2. The results indicate

substantial errors in the RRM-PD simulated aerosol sources, sinks, and their components, which are consistent with the spatial distribution of relative differences shown in Supplementary Fig. S3-S9.

The N_RMSE for both BC and POM total sources is approximately 71%. However, the error contributions from anthropogenic sources are much larger for BC, with a contribution of about 49%, compared to POM, which has a contribution of about 16%. POM errors are primarily driven by the biomass-burning sources. This difference is due to the spatial variability

and magnitudes of biomass burning emissions (BB) in 2014, which are reflected in the surface and elevated emission ratios of BC and POM. It should be noted that the contribution from BB sources (e.g., elevated emission) is comparatively significant over NA due to the 2014 Northwest Territories fires. For BC, larger BB sources are concentrated within Northwest NA, while anthropogenic sources are prevalent over the rest of NA (Supplementary Fig. S3). In contrast, for POM, the BB sources has significantly larger contribution and the large anthropogenic sources are concentrated over the eastern and southern NA (Fig.

S5). These source contributions are reflected into the spatial distribution and magnitude of relative differences, resulting in an uneven error contribution to BC and POM from different emission sources.

Total simulated aerosol sink yields a N_RMSE of 45% and 34% from BC and POM respectively, with the largest contributions from dry deposition (Fig. 10). This is consistent with the spatial distribution of relative differences over NA between RRM-SE-PD and RRM-PD in Fig. S4 and S6. Aerosol dry deposition in EAMv2 depends on gravitational settling and turbu-

lent deposition velocities, and at the surface turbulent deposition is substantially larger for BC and POM. Our results show that over 99% of the N_RMSE in dry deposition comes from turbulent dry deposition flux, which is consistent with the fractional distribution of sinks in Fig. 9.

N_RMSE for wet deposition components for POM and BC are significantly smaller than dry deposition, which is inconsistent to the fractional contribution of sinks. EAMv2 aerosol wet deposition treatment considers both in-cloud and below-cloud

scavenging by stratiform and convective precipitation Barth et al. (2000); Rasch et al. (2000). In-cloud scavenging is the dominant driver of total wet deposition, so we focus on the stratiform and convective in-cloud scavenging analysis in EAMv2. In-cloud scavenging through stratiform clouds considers only the cloud-borne aerosol particles, while in convective clouds both the cloud-borne and interstitial aerosols are affected. For convective in-cloud scavenging, EAM uses a unified convective transport/removal treatment, which considers secondary activation for aerosols into convective updrafts and a subsequent

in-cloud wet removal (Wang et al., 2020, 2013). BC and POM in the primary carbon mode are not directly affected by the convective in-cloud scavenging. However, they can be converted to the accumulation mode particles through condensation-aging and coagulation, which are strongly affected by the convective removal. Similarly, stratiform in-cloud scavenging strongly affects stratiform cloud-borne BC and POM in the accumulation and coarse modes, which are formed via droplet nucleation. Since neither of the in-cloud scavenging routines directly affects the prescribed BC and POM in primary carbon mode, we

observe a weaker error contribution from wet deposition, despite its strong contribution to the total aerosol sinks.





For sulfate aerosols, the simulated source and sink yield a N_RMSE of ~36% and ~9% respectively (Fig. 11). The error estimates for sources align with the fractional distribution of each component, with the largest contributions from gas-aerosol exchange and in-cloud aqueous-phase ($SO_4$) chemistry. Sulfate sinks yield significantly smaller errors compared to BC and POM, consistent with the weaker contribution from dry deposition. Interestingly, we see larger error contributions from wet deposition, with stratiform in-cloud scavenging accounting for more than half of it. Prescribed sulfate aerosol emissions are in the Aitken and accumulation modes, which are hygroscopic and can be directly affected by convective in-cloud scavenging. On the other hand, the sulfate production through in-cloud aqueous-phase chemistry is attributed to cloud-borne aerosol particles in stratiform clouds. These particles are subsequently removed through stratiform in-cloud scavenging, which is the largest contributor to total wet deposition. The error estimates for each sink component are consistent with their fractional contributions to the total simulated sulfate sink. Therefore, we see larger N_RMSE from in-cloud wet depositions for sulfate aerosols.

Overall, our analysis highlights significant errors in simulated aerosol sources and sinks when the default emission treatment is used. Larger simulation errors are evident for BC and POM source-sink components, primarily due to the prescribed emissions and dry depositions.

### 3.3 Model evaluation against observations

To demonstrate the improved heterogeneity resulting from the revised emission treatment, ideally it would be nice to evaluate the model against observations near locations with sharp emission gradient, where the linear interpolation of coarse-resolution data causes large errors. The sharpest gradients in BC emissions are found over the northeast US (see emission difference shown in Fig. S1a), while for central and western US, the emission spatial gradient is weaker. Consequently, many observational network sites (e.g., IMPROVE and AERONET, which mainly represent regional background conditions) do not fall within or near the regions with large errors.

To address this issue, we have applied conditional sampling based on the emission differences between the new and default treatment. Specifically, we select sites where the emission differences are above (below) the 75th (25th) percentiles. Using the 90th/10th percentiles yields similar results, but with a reduced number of sites. Without conditional sampling, sites falling within the 75th/25th percentiles can mask the impact of the revised treatment (not shown). Conditional sampling raises the likelihood of selecting sites over large error regions, assuming larger gradients occur near larger biases.

In Fig. 12, we evaluate the simulated BC and POM surface concentrations in RRM-PD and RRM-SE-PD against observations from the IMPROVE (described in section 2.5). A similar evaluation for LR simulations is shown in Fig. S10. Figure 12 shows substantial improvements in simulated surface concentrations of BC and POM with RRM-SE-PD. The spatial correlations between the simulated and observed surface concentrations are increased from 0.44 to 0.59 for BC and 0.43 to 0.51 for POM, when the revised emission treatment (RRM-SE-PD) is applied. Our results suggest significant improvements in spatial heterogeneity in terms of the spatial correlation coefficient for BC (with Fisher's Z of -1.8 at $p \leq 0.05$) and potential improvements for POM (with Fisher's Z of -1.2 at $p \leq 0.1$).

In Fig. 13, we present the distribution of daily mean surface concentrations of BC and POM using the conditionally sampled sites (as in Fig. 12). It shows that RRM-PD consistently underestimates the seasonal means and the variability of observed daily





mean measurements. This underestimated variance can be attributed to an inaccurate representation of spatial heterogeneity in the default emission treatment. RRM-SE-PD reduces the seasonal mean biases along with improved variability in simulated daily mean surface concentration. We note that using emissions from the year 2016 (which are unavailable for EAMv2) in the simulations will provide more accurate comparisons (as discussed in section 2.4). However, the overall changes in global emissions are small (∼10%) and are unlikely to significantly affect the simulated surface concentrations. For example, the total

anthropogenic BC emissions over North America decreased from 0.3 Tg/yr in 2014 to 0.28 Tg/yr in 2016 (around 7% change) in the updated CEDS version (McDuffie et al., 2020). Therefore, the small year−to−year variation from 2014 to 2016 is not expected to have a large impact on the bias estimates for BC.

The spatial correlations (e.g., ∼0.72) between simulated and observed sulfate surface concentrations show no significant improvement in RRM-SE-PD compared to RRM-PD. This is likely due to the fact that most of the sulfur emissions are not

distributed to the surface layer. About 80% of the total sulfur emissions prescribed in EAMv2 are from elevated sources, mainly industrial and energy sources (∼95%) and are prescribed at 100-300 meters above the surface. Large differences exist for elevated sources as opposed to the surface sources (Fig. S1 c, d). Less than 20% of the total sulfate and $SO_2$ emissions over North America (primarily from transportation, and residential sectors) are emitted to the surface layer that will immediately affect the surface concentration. This leads to a weaker sensitivity to emission changes in simulating sulfate surface concentration.

EAMv2 overestimates sulfate ($SO_4$) aerosol surface concentrations for all simulations with a normalized bias exceeding 130% (Fig. 14). Although the sulfur emission is increased by ∼18% from 2014 and 2016, this can not explain the large overestimation in the surface $SO_4$ concentration.

AOD does not show any improvements either, with a spatial correlation of ∼0.35 for both RRM-PD and RRM-SE-PD (Fig. 14b, d). Since AOD is the column-integrated aerosol extinction, it is strongly modulated by other processes, such as chemistry,

horizontal transport, removal, and vertical diffusion. However, significant differences may exist in aerosol extinction near the surface layers between RRM-PD and RRM-SE-PD simulations (as shown in section 3.2.1), where the emission changes have larger impact.

Emissions in LR-PD and LR-SE-PD have similar resolutions and their difference (in terms of normalized RMSE, Table 1) is much smaller compared to the difference between RRM-PD and RRM-SE-PD (Table S1). Therefore, the LR-PD and LR-SE-

PD simulations did not show significant differences in surface concentrations or AOD (as shown in supplementary Fig. S10 and S11).

Overall, our results suggest that the revised emission treatment can significantly improve the spatial heterogeneity and daily variability in magnitudes of simulated surface concentration of aerosol species that are primarily emitted from surface in high-resolution simulations. More evaluations against observations near polluted (e.g., urban) areas will be helpful in future

high-resolution model applications.

### 3.4 Impact on the anthropogenic aerosol forcing

As shown in section 3.2, there are large changes in lower-tropospheric aerosol mass concentration and optical properties. Here we evaluate whether the revised emission treatment will affect the anthropogenic aerosol forcing estimate. We calculate



the anthropogenic aerosol forcing by taking the difference between simulations with present-day (PD) and pre-industrial (PI)
emissions of aerosols and precursors (see section 2.4). We further decompose the aerosol forcing ($\Delta F$) following Ghan (2013)
to estimate the forcing caused by aerosol-radiation interaction (or, direct aerosol effect, DIR), aerosol-cloud interaction (mainly
from indirect aerosol effect, IND), and other residual factors (RES, primarily surface albedo changes).

Supplementary Fig. S12 shows the spatial distribution of decomposed Top-Of-Atmosphere (TOA) $\Delta F$, calculated for the
default emission treatment. Over North America (NA), the net aerosol forcing ($\Delta F$) is about -2.4 W m$^{-2}$ (Fig. S13a), with
the largest contribution from the indirect aerosol effect (IND, Fig. S13d), particularly the shortwave component ($\Delta F_{SW}$, Fig.
S13e). Net longwave aerosol forcing is positive (0.487 W m$^{-2}$, Fig. S13c), which is mostly driven by the indirect aerosol effect
over ocean (Fig. S13f) and the residual effect over land (Fig. S13o). The direct aerosol effect is very small (-0.035 W m$^{-2}$,
Fig. S13g).

Figure S13 illustrates the difference in decomposed forcing estimates between the revised and default emission treatments.
There are relatively large differences between the original and revised treatments, in terms of RMSE. Net aerosol forcing over
NA (Fig. S14a) has an RMSE of 1.104 W m$^{-2}$, which is mostly driven by the indirect aerosol effect (Figure S14d). Although
there are large local differences between the two emission treatments (N_RMSE of 49% in the estimated net aerosol forcing
over NA), these differences are mostly caused by perturbations (noises) in the cloud fields and the large differences are mostly
away from aerosol emission sources. In contrast, the relative difference in the annual mean anthropogenic aerosol forcing
estimate over NA is only about 3-5% between the revised and default emission treatments. Therefore, for the anthropogenic
aerosol forcing estimate, the overall impact of the revised emission treatment is small.

## 4    Conclusions

In this study, we improve the emission treatment in E3SM to better represent the anthropogenic aerosol emission in the simu-
lations. The default E3SM emissions treatment fails to preserve spatial heterogeneity or conserve mass, resulting in substantial
errors. The improved treatment accurately preserves the original spatial heterogeneity both inland and near the coast, providing
a more accurate representation of surface and vertically distributed emissions. Our results indicate that the default treatment for
anthropogenic aerosol emissions in both LR and RRM simulations consistently yields large root mean square errors (RMSE)
for all species. The improved emissions data prepared for the revised emission treatment can resolve these issues by maintain-
ing spatial heterogeneity and mass conservation. The revised treatment is also applicable to the emission input of other species
at different model resolutions.

We find significant differences in the simulated aerosol surface concentration between the default and revised emission treat-
ments. The differences are closely linked to the spatial distribution of prescribed emission gradients and they persist at higher
elevations above the surface. Larger discrepancies are observed in high-frequency concentration profiles. This suggests that
errors in default emissions can have a significant impact on the simulated aerosol properties, especially at high resolutions.
Consistent with the near surface mass concentration changes, the default treatment also yields larger errors in simulated extinc-
tion and absorption profiles near and above the surface, leading to significant errors in high-frequency aerosol optical depths.



This result is particularly relevant, given that short-term high-frequency fields are often used for model evaluations against observations.

Using the default emission treatment can lead to significant errors in aerosol sources and sinks. Substantial errors in the simulated aerosol sources, sinks, and their decomposed components are found along with large relative differences in their spatial distribution. The prescribed emissions and dry deposition are the dominating error terms in BC and POM source-sink components. Errors for wet deposition components for POM and BC were significantly smaller than dry deposition, despite the large fractional contribution to sink. This is mainly due to the fact that in-cloud scavenging processes do not directly affect prescribed BC and POM in the hydrophobic primary-carbon mode upon emission. The largest error contributions for sulfate

aerosol sources come from gas-aerosol exchange and in-cloud aqueous-phase chemistry, while wet deposition, particularly stratiform in-cloud scavenging, is the biggest contributor to errors in sinks.

The revised emission treatment leads to improved heterogeneity in simulated surface concentration, particularly in regions with sharp emission gradients. Furthermore, simulations with the revised treatment consistently outperformed simulations with the default treatment in terms of seasonal mean biases and variability of daily mean surface concentrations of black carbon and

primary organic matter. Our study highlights the importance of maintaining spatial heterogeneity in high-resolution simulations for aerosol and Aerosol−Cloud Interactions (ACI) studies.

*Code and data availability.*    The original E3SMv2 source code is available at https://doi.org/10.11578/E3SM/dc.20210927.1 (E3SM Project, 2021). The modified source code for the revised emission treatment is available at https://doi.org/10.5281/zenodo.7823633 (Hassan et al., 2023). The model-native grid anthropogenic aerosol emissions data used in this study are available at https://doi.org/10.5281/zenodo.7823686

(Hassan and Zhang, 2023). The ERA5 data can be obtained from Copernicus Climate Change Service Climate Data Store (CDS), at https://doi.org/10.24381/cds.6860a573 (Hersbach et al., 2020). The source code of the python package used for remapping emissions is available at https://doi.org/10.5281/zenodo.7931486 (Hassan, 2023)

*Author contributions.*    TH and KZ designed the study and developed the new emission treatment in E3SM with help from BS. TH designed the Python package and conducted the simulations. TH prepared the ERA5 nudging data with instructions from SZ and KZ. TH processed

the data and performed the analysis with discussion with KZ, JL, HW, and PM. TH wrote the original manuscript; all authors reviewed and edited the manuscript.

*Competing interests.*    One of the (co-)authors is a member of the editorial board of Geoscientific Model Development. Other authors have no other competing interests to declare.



*Acknowledgements.* This work is supported by the Enabling Aerosol–cloud interactions at GLobal convection-permitting scalES (EAGLES)

project (project no. 74358). The EAGLES project is sponsored by the U.S. Department of Energy, Office of Science, Office of Biological and Environmental Research, Earth System Model Development (ESMD) program area. HW also acknowledges support by the Energy Exascale Earth System Model (E3SM) project. The Pacific Northwest National Laboratory (PNNL) is operated for the DOE by the Battelle Memorial Institute under Contract DE-AC05-76RL01830. This research used high-performance computing resources from the PNNL Research Computing and resources of the National Energy Research Scientific Computing Center (NERSC), operated under Contract No.

DE-AC02-05CH11231.



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





**Table 1.** List of simulations performed and analyzed in this study. All simulations are nudged towards the ERA5 reanalysis. Low resolution (LR) "ne30pg2" resolution refers to a dynamics grid spacing of ~110 km (~1°). High resolution mesh (dynamics grid spacing of ~28 km) is over the North America in Regionally Refined Model (RRM) simulations. RLL refers to the Regular latitude/longitude grid. SE refers to the new emission treatment. Present-day (PD) and pre-industrial (PI) simulations are conducted with anthropogenic aerosol emissions from year 2014 and 1850 respectively.

| Group | Simulation name | Model Resolution | Resolution of emission data | Remapping method |
|-------|-----------------|------------------|------------------------------|------------------|
| 1 | LR-PD (PI) | ne30pg2 | ~2° RLL | Linear interpolation |
|   | LR-SE-PD (PI) | ne30pg2 | ne30pg2 | Conservative remapping |
| 2 | RRM-PD (PI) | NA RRM | ~2° RLL | Linear interpolation |
|   | RRM-SE-PD (PI) | NA RRM | NA RRM | Conservative remapping |



**Table 2.** EAMv2 anthropogenic aerosol emissions data statistics in the default emission treatment for present-day (PD) RRM simulations. Statistics are shown for both the surface and elevated emissions of different aerosol species. All estimates are over the North America land surface. Mean values indicate the area weighted mean emission fluxes. NMB, NStdDevB, and N_RMSE are defined as $\left(\frac{\sum(emis_{lin}-emis_{accurate})}{\sum emis_{accurate}}\right) \times 100\%$, $\frac{stdDev_{lin}-stdDev_{accurate}}{stdDev_{accurate}}$, $\frac{RMSE}{stdDev_{accurate}} \times 100\%$ respectively. The "accurate" subscript indicates data that preserve spatial heterogeneity and conserve mass. The "lin" subscript indicates linearly interpolated data used in the default treatment. Units of Mean, StdDev, and RMSE are in kg/m$^2$/s. N_RMSE and NMB are in percentage (%). NStdDevB is unitless.

| Aerosol | Emission space | Mean [$\times 10^{-12}$ kg/m$^2$/s] (accurate) | NMB [%] | StdDev [$\times 10^{-12}$ kg/m$^2$/s] (accurate) | NStdDevB | RMSE [$\times 10^{-12}$ kg/m$^2$/s] | N_RMSE [%] |
|---|---|---|---|---|---|---|---|
| BC | surface | 6 | 8.398 | 10 | -0.395 | 9 | 67 |
| | elevated | 2 | 2.606 | 20 | -0.423 | 10 | 72 |
| POM | surface | 20 | 9.899 | 50 | -0.369 | 30 | 62 |
| | elevated | 50 | 1.270 | 500 | -0.422 | 400 | 71 |
| SO$_4$ | surface | 0.6 | 1.085 | 1 | -0.251 | 0.7 | 54 |
| | elevated | 5 | 5.648 | 20 | -0.525 | 20 | 84 |



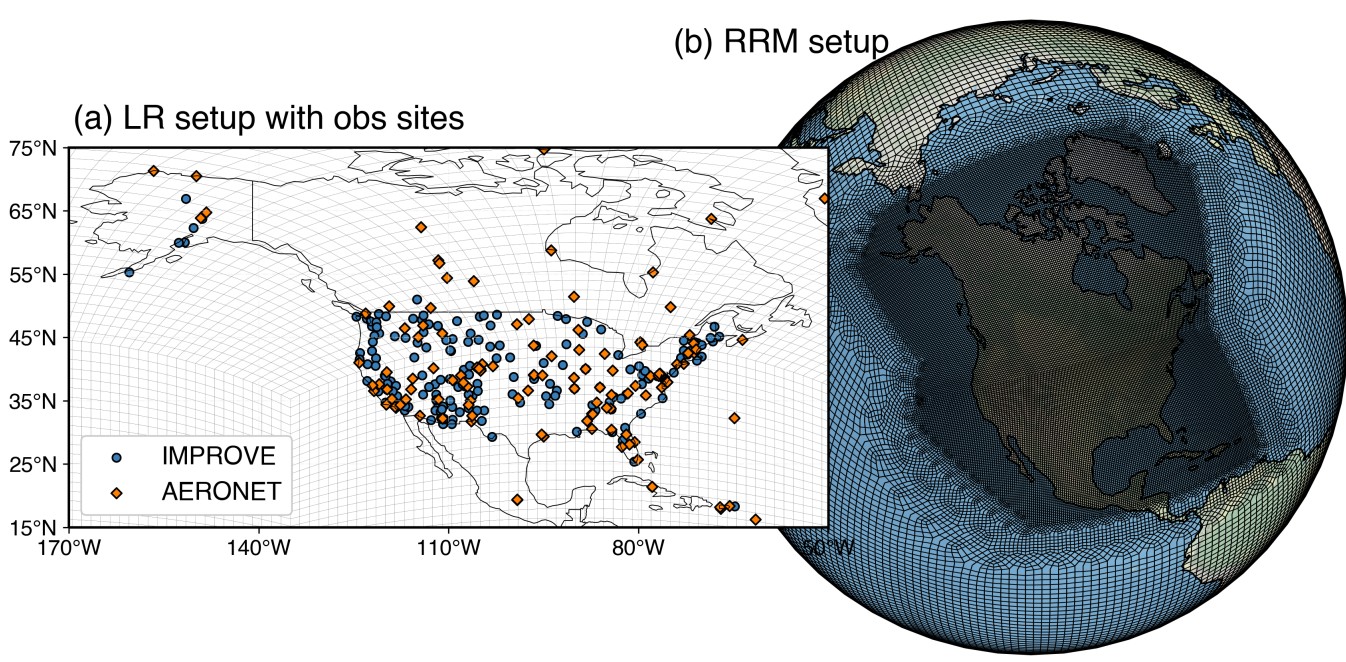

**Figure 1.** The model-native grid configuration for (a) low-resolution, LR (ne30pg2) grid, and (b) regionally refined mesh over North America (NA RRM) are shown. The locations for point-source aerosol measurement sites over North America from IMPROVE and AERONET sites are also shown overlaid on the LR grid in panel (a).



(a) Emission Treatment in EAMv2  (b) Default vs New Emission Treatment

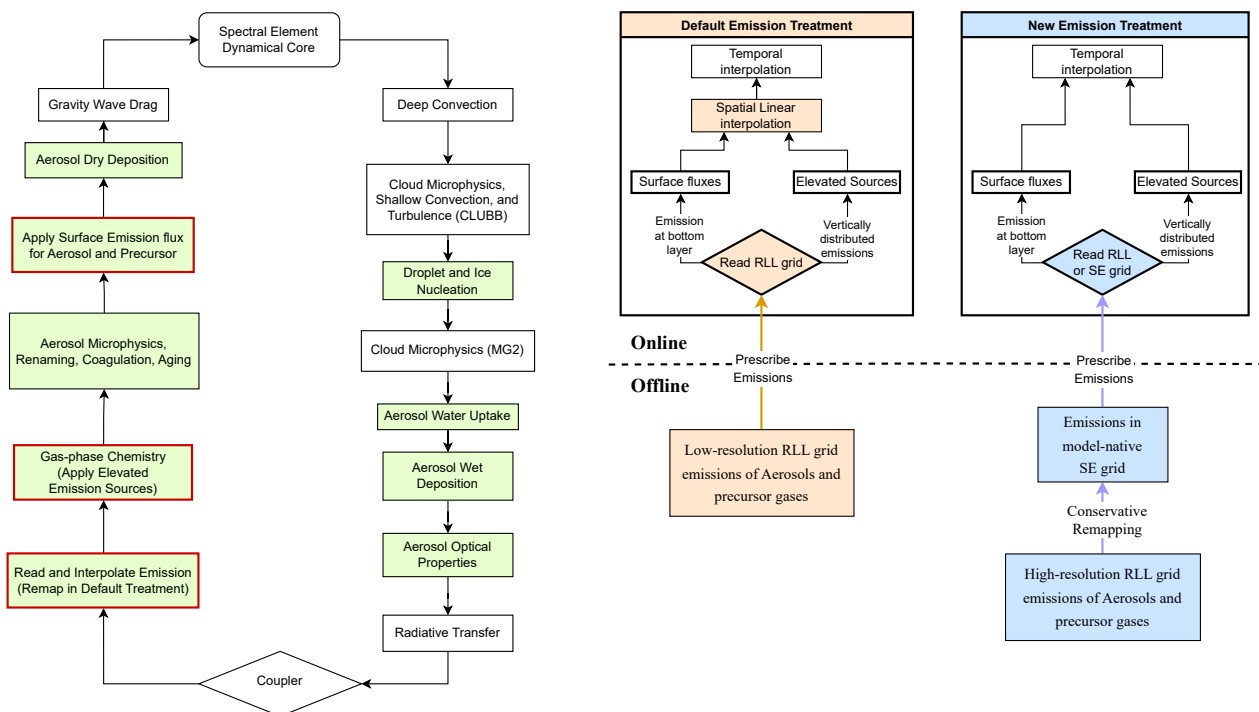

**Figure 2.** Flowcharts illustrating how the aerosols and precursor gas emission is handled in the EAMv2 simulation. Light green boxes in panel (a) depicts the aerosol-related processes, with red bordered boxes indicating emission-related modules impacted during one time step of the physics and dynamics calculations. Panel (b) compares the default and the new emission treatment. The key differences are depicted by light orange and light blue boxes for default and new emission treatment respectively.

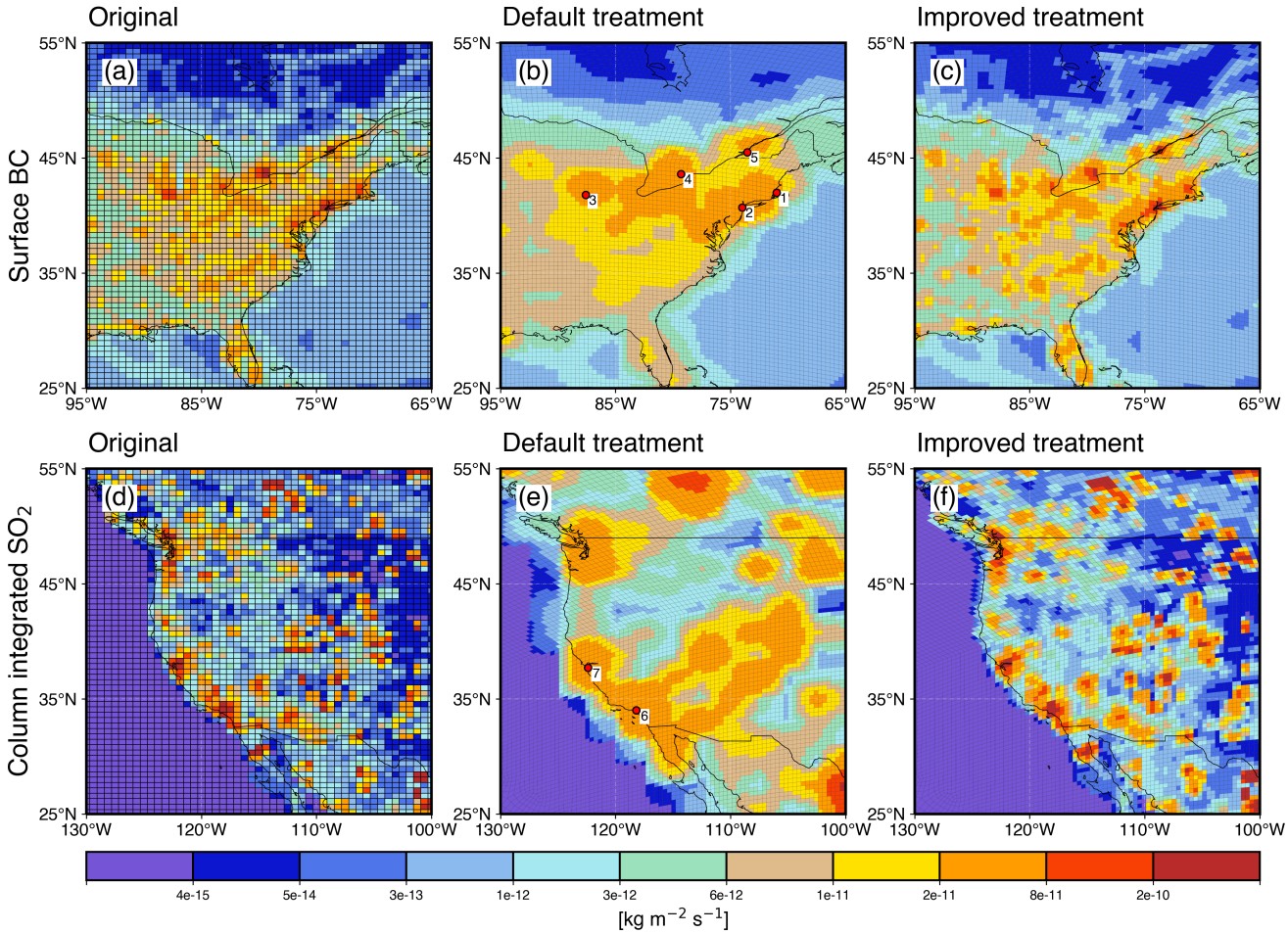

**Figure 3.** Spatial distribution of the present-day surface BC emissions (top) and column integrated $SO_2$ emissions (bottom) from the original high-resolution data (a, d), the default emission treatment (b, e), and the improved emission treatment (c, f) for RRM simulations. Distributions are shown over eastern (western) United States for BC ($SO_2$) emissions in kg/m$^2$/s units. Red circles in panel b and e indicate major cities with large anthropogenic BC and $SO_2$ emissions respectively. Markers titled 1, 2, 3, 4, 5, 6 and 7 depict Boston, New York, Chicago, Toronto, Montreal, Los Angeles, and San Francisco respectively.





**Figure 4.** Simulated spatial distribution of annual mean aerosol surface concentration from RRM-PD (left column) and the relative difference between RRM-SE-PD and RRM-PD (right column) over North America. Distributions are shown for (a, b) Black Carbon (BC), (c, d) Primary Organic Matter (POM), and (e, f) Sulfate aerosols. The relative difference for field X is calculated as: $\left(\frac{X_{se}-X_{def}}{X_{def}}\right) \times 100\%$, where "se" and "def" subscripts refer to the simulations with new and default emission treatment respectively. Mean, RMSE and normalized RMSE (N_RMSE) are indicated at the top right corner of the panels. Mean and RMSE has a unit of $\mu g\ m^{-3}$. N_RMSE is defined as in Table 2.





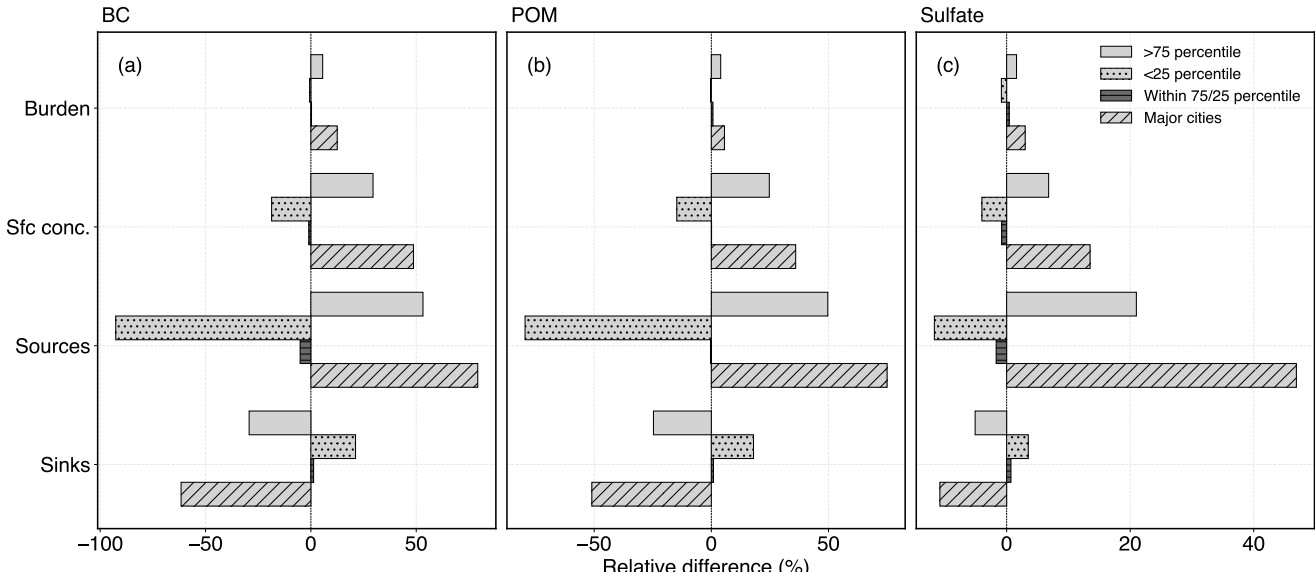

**Figure 5.** Quantitative distribution of annual mean relative differences between RRM-SE-PD and RRM-PD for simulated aerosol burden, surface concentration, aerosol sources and aerosol sinks. Distributions are shown for (a) BC, (b) POM, and (c) sulfate aerosols in percentage. Simulated fields are masked by emission differences between the new and default treatment. Masking is applied for regions above 75th percentile (>75th percentile), below 25th percentile (<25th percentile), and within 25-75th percentiles over North America. Urban-scale differences are also shown considering major cities with larger anthropogenic aerosol emissions.







**Figure 6.** Spatial distribution of annual mean simulated (a, b) aerosol extinction at surface, (c, d) aerosol absorption at surface, (e, f) Aerosol Optical Depth (AOD), and (g, h) absorbing AOD from RRM-PD (left column) and the relative difference between RRM-SE-PD and RRM-PD (right column) over North America. The relative difference for field X is calculated as: $\left(\frac{X_{se}-X_{def}}{X_{def}}\right) \times 100\%$, where "se" and "def" subscripts refer to the simulations with new and default emission treatment respectively. Mean, RMSE and normalized RMSE (N_RMSE) are indicated at the top right corner of the panels. Mean and RMSE has a unit of $\mu g\ m^{-3}$. N_RMSE is defined as in Table 2.





**Figure 7.** Daily mean Black carbon (BC) concentration profile and burden time-series during the month of July of 2016 near sharp spatial emission gradient in eastern North America. Simulated vertical distribution and burden time-series from RRM-PD (left column) and the relative difference between RRM-SE-PD and RRM-PD (right column) are shown. Panels a-d depicts highly polluted location, with panels e-h depicting nearby less polluted location.





**Figure 8.** Daily mean aerosol extinction and absorption time-series during the month of July of 2016 over highly polluted location as in Fig. 7 near sharp spatial emission gradient in eastern North America. Simulated aerosol extinction profile, absorption profile, AOD, and absorption AOD (AAOD) time-series from RRM-PD (left column) and the relative difference between RRM-SE-PD and RRM-PD (right column) are shown.



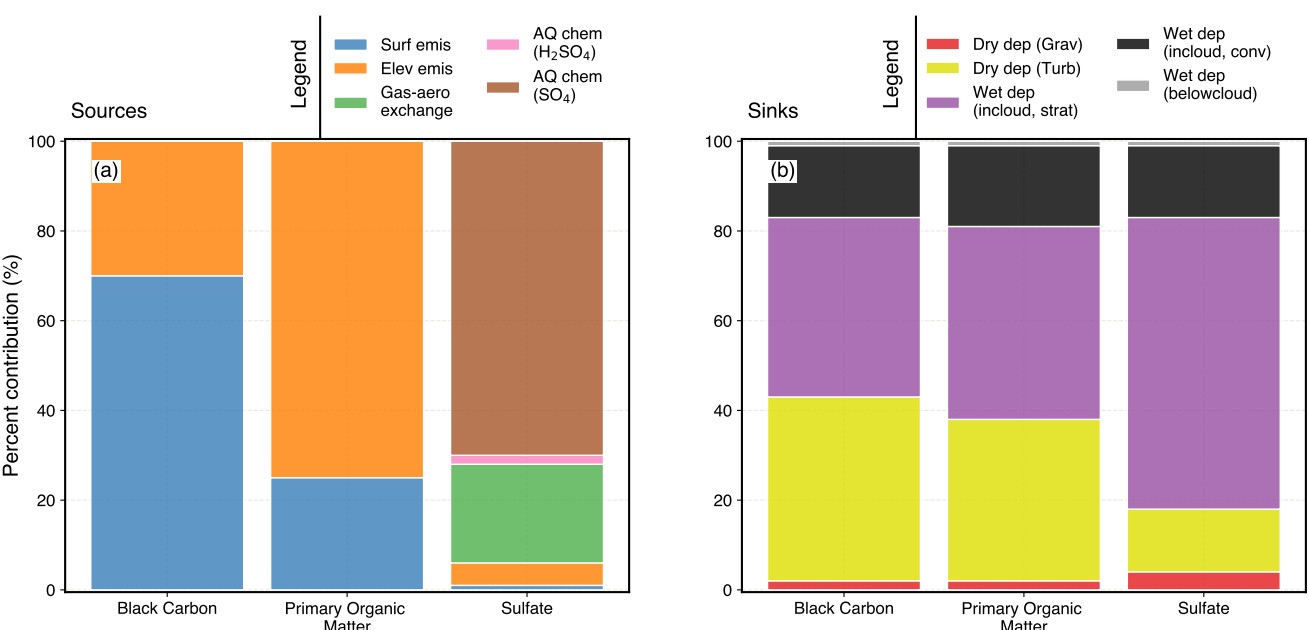

**Figure 9.** Contribution of the decomposed aerosol source and sink processes in EAMv2 simulations for Black Carbon (BC), Primary Organic Matter (POM), and Sulfate (SO$_4$) aerosols. All estimates are from annual means over North America. Legend for sources and sinks are shown separately at the top right corner of each panel. Decomposed dry deposition processes "Grav" and "Turb" refers to the gravitational settling and turbulent deposition respectively. "AQ chem" refers to chemical production through in-cloud aqueous chemistry.



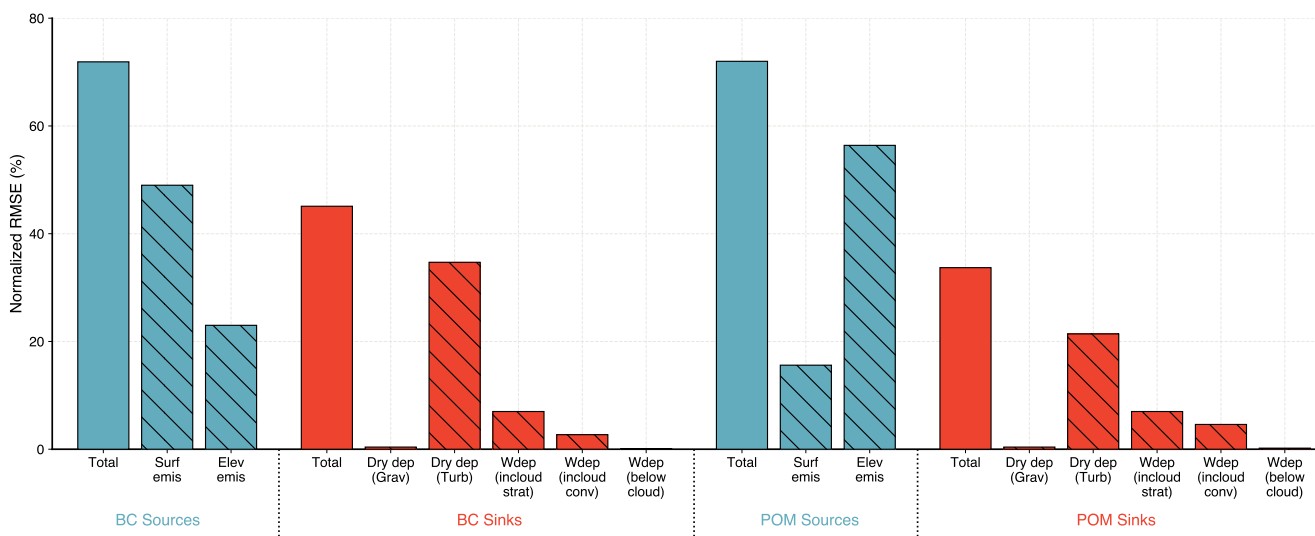

**Figure 10.** Error estimates of the total and decomposed processes contributing to the simulated sources and sinks of BC and POM from present-day (PD) RRM simulations. All RMSE estimates are over the North America land surface. Normalized RMSE is defined as $N\_RMSE_{Total} \times \frac{Weight_i \times N\_RMSE_i}{\sum (Weight_i \times N\_RMSE_i)}$ , where N_RMSE is defined as in Table 2, weights are from Figure 6, and subscript "i" indicates decomposed process. The solid bars represent the normalized RMSE for the total source and sink. The hatched bars are the decomposed processes normalized to the total N_RMSE values. The "Wdep" bars refer to the wet deposition components.



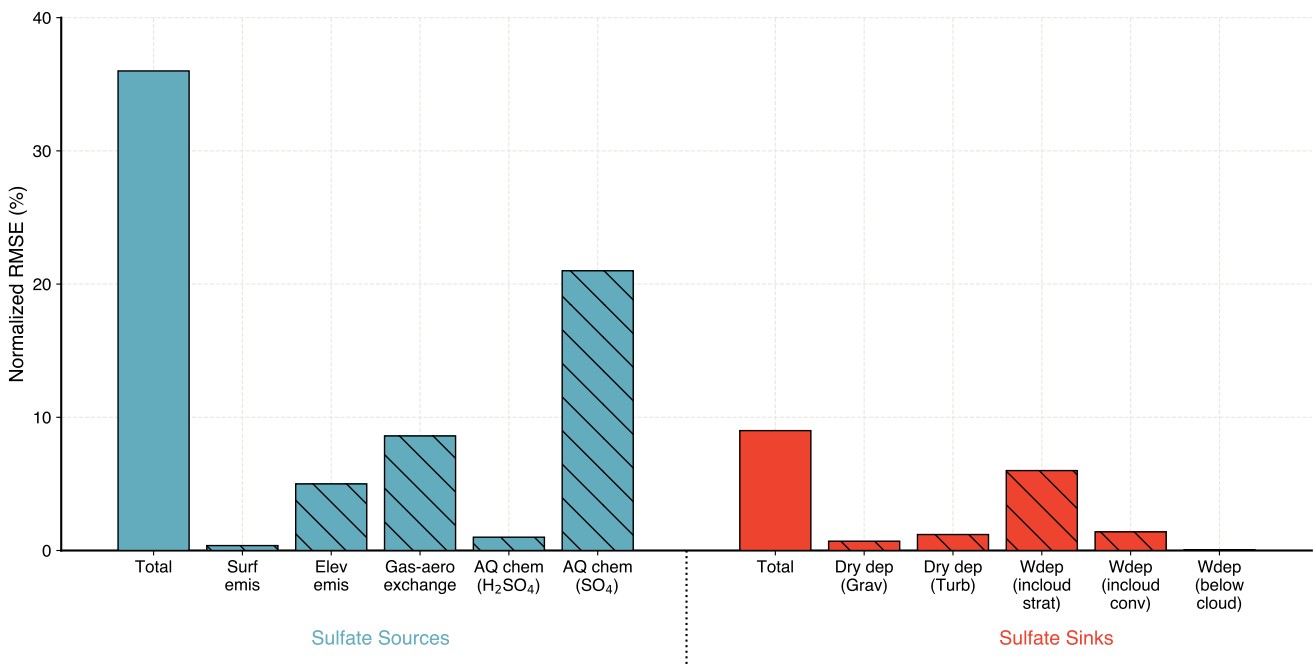

**Figure 11.** Error estimates of the total and decomposed processes contributing to the simulated sources and sinks of Sulfate aerosols from present-day (PD) RRM simulations. All RMSE estimates are over the North America land surface. Normalized RMSE is defined as $N\_RMSE_{Total} \times \frac{Weight_i \times N\_RMSE_i}{\sum (Weight_i \times N\_RMSE_i)}$ , where N_RMSE is defined as in Table 2, weights are from Figure 6, and subscript "i" indicates decomposed process. The solid bars represent the normalized RMSE for the total source and sink. The hatched bars are the decomposed processes normalized to the total N_RMSE values. The "Wdep" bars refer to the wet deposition components.






**Figure 12.** Scatter plots between simulated and observed monthly mean surface concentrations of (a, c) Black Carbon (BC) and (b, d) Primary Organic Matter (POM). Observations of the surface concentrations are from IMPROVE for the simulation year of 2016. Scatter plot statistics compare the spearman's correlation (R), number of data points (n), RMSE, NMB values between (a, b) RRM-PD and (c, d) RRM-SE-PD simulation. RMSE and NMB are defined as in Table 2. Solid lines indicate the 1:1 ratio, and the dashed lines indicate the 1:2 and 2:1 ratio. The values at the top of each column indicate the observed mean.



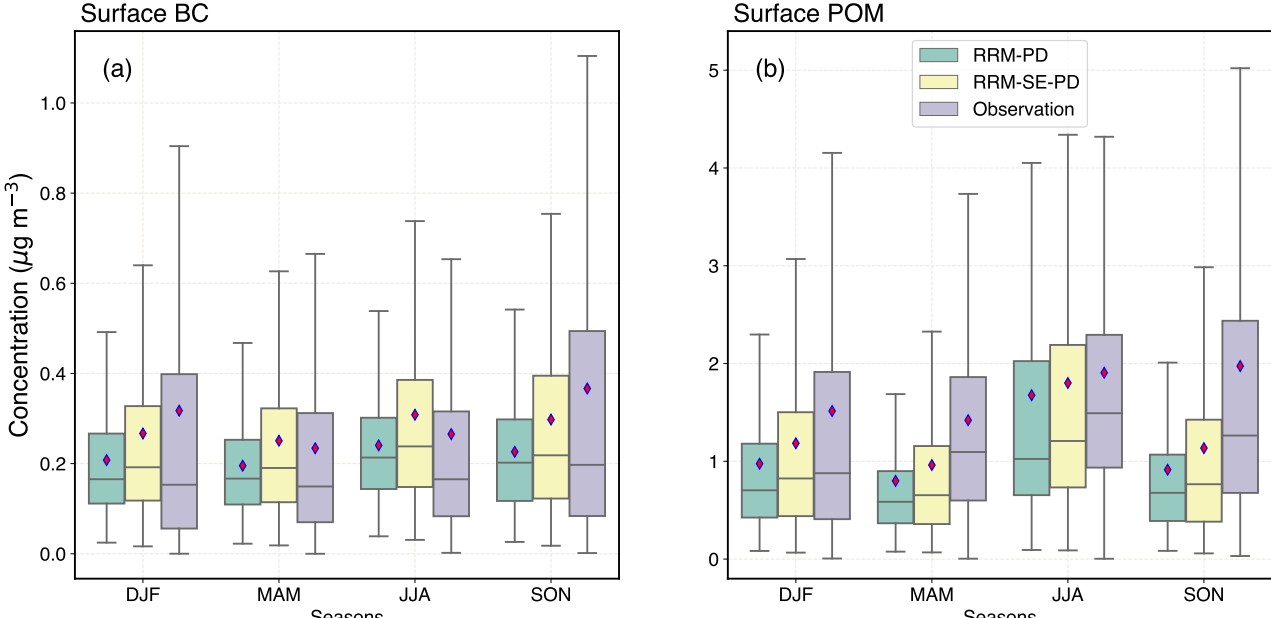

**Figure 13.** Boxplot comparison of the daily mean distribution for (a) BC and (b) POM surface concentrations from RRM-PD simulation, RRM-SE-PD simulation and IMPROVE network measurements. The whiskers are based on 1.5 times interquartile range (IQR). Distributions are plotted for different seasons over the simulation year, with red diamonds indicating the seasonal means.





**Figure 14.** Scatter plots between simulated and observed monthly mean surface concentrations of (a, c) sulfate ($SO_4$) aerosols and (b, d) Aerosol Optical Depth (AOD) at 550 nm. Observations of the surface concentrations and AOD are from IMPROVE and AERONET respectively for the simulation year of 2016. Scatter plot statistics compare the spearman's correlation (R), number of data points (n), RMSE, NMB values between (a, b) RRM-PD and (c, d) RRM-SE-PD simulation. RMSE and NMB are defined as in Table 2. Solid lines indicate the 1:1 ratio, and the dashed lines indicate the 1:2 and 2:1 ratio. The values at the top of each column indicate the observed mean.