# Peer review of "Impacts of spatial heterogeneity of anthropogenic aerosol emissions in a regionally-refined global aerosol-climate model"

_EGUsphere, 2023_

## Referee Comment (RC1)

**Review of GMD-2023-1055 « **Impacts of spatial heterogeneity of anthropogenic aerosol emissions in a regionally-refined global aerosol-climate model"**

In this article, the authors update the pre processing and in particular the remapping of emission datasets from the original resolution to the model resolution. This is a problem faced by all atmospheric composition systems and as such, there is a possible interest from the community in such aspects. The paper is well written and organized and the plots are nicely done. However, I feel that the comparison made in the paper is unfair: it is not a surprise that gradients are better represented with emissions at a resolution corresponding to ~42km grid than with emissions at 1.9x2.5° resolution. It is very possible that I misunderstood something, but why didn't the authors use the original CEDS or CMIP6 emissions at 0.25x0.25° or 0.5x0.5° as an input to the original emissions treatment (left side of Figure 2b) instead of the 1.9x2.5° resolution? Comparing simulations with the old emissions treatment and this high resolution input with simulations with the new emissions treatment using the same input would be more meaningful, and more interesting for the reader. As such, I would recommend a major revision, and suggest to the authors that they rewrite their manuscript in order to show the added value of their new approach but using the **same** emission datasets as input for the two simulations. Otherwise it is hard to discriminate between the added value of the new emissions treatment and that of using higher resolution emissions (which is well known). A side topic of the paper could be the mass conservative aspect of their revised treatment of aerosol emissions – how much does it change emissions as compared to the (I suppose) non mass conservative remapping/interpolation used before? With what impacts on simulated aerosol burden/surface concentration? I think also more detail should be given as to how/with which method is mass conserved in the revised treatment.

---

## Referee Comment (RC2)

**Review of "Impacts of spatial heterogeneity of anthropogenic aerosol emissions in a regionally-refined global aerosol-climate model" by Hassan et al.**

**General comments:**

This work implements a revised emission treatment in E3SM to preserve the original emission spatial heterogeneity and conserve emission mass fluxes in the simulations. The authors show significant differences in the simulated surface concentration of aerosols between the default and revised emission treatments in regionally-refined high-resolution simulations. They also show that the revised emission treatment leads to improved heterogeneity in simulated surface concentration of aerosols, particularly in regions with sharp emission gradients.

This study is interesting and the subject is of great interest to GMD. The manuscript is well written. However, the interpretations of the simulations with low-resolution emissions and the comparisons are not satisfactory, which are cause for concern (see Major comments). Most other comments listed below are minor clarifications. Once these points are addressed satisfactory, the paper should in my opinion be suitable for publication in GMD.

**Major comments:**

I cannot understand the author's intention about the simulations and comparisons. I may misunderstand something, but I give comments here.

I think that it is common to use the original emission data for model simulations. I cannot understand why the authors do not use the original CMIP6 emissions (CEDS 0.5° x 0.5° and GFED4 0.25° x 0.25°) as inputs to the default emission treatments (LR-PD), instead of the default low-resolution (1.9° x 2.5°) prescribed emissions. Is this a problem specific to the model used in this study?

I cannot understand why the authors conduct high-resolution (42 km) simulation (RRM-PD) with much low-resolution emission (1.9° x 2.5°). Because there is a large difference in the resolutions between two (42 km and 1.9° x 2.5°), it would be qualitatively obvious that high-resolution simulations with low-resolution emission cannot reproduce aerosol concentrations at the surface in highly polluted regions. Even though the authors understand these things, do they have some other purpose in

performing this simulation, such as an evaluation of the impacts in advance before doing cloud-resolving scale simulations with relatively low-resolution (0.25°-0.5°) emissions (although the evaluation would be difficult)?

Although the authors basically compare RRM-PD and RRM-SE-PD simulations or compare LR-PD and LR-SE-PD simulations, comparison between LR-SE-PD and RRM-SE-PD simulations (using the same original CMIP6 emissions?) would be meaningful. For example, there is a better agreement between simulated and observed BC and POM in RRM-SE-PD (Figure 12c-d) than those in LR-SE-PD (Figure S10c-d).

If the authors want to show influences of improvement of the heterogeneity and mass conservation separately, additional low-resolution (165 km) simulation and high-resolution (42 km) simulation using the same original CMIP6 emissions as inputs to the default emission treatments and comparisons with other four simulations (shown in Table 1) would be helpful. However, because comparison between LR-PD and LR-SE-PD simulations does not show significant difference in surface aerosol concentrations and AOD (Lines 443-446, Figures S10-S11), it may be difficult to evaluate the quantitative separation.

The authors should explain the objectives of the simulations and their comparisons in more details. It is unclear to me.

The authors point out the problem of mass conservation in the default emission treatment, however, it seems that there is almost no description about them. Could the authors add explanations about influences of the mass conservation?

**Specific comments:**

Lines 165-171, please describe the horizontal resolution of the reanalysis data used for the nudging.

Lines 261-264 and 428-432, the authors state that the small difference in sulfate aerosols is attributed to the fact that prescribed sulfur emissions are mostly emitted from elevated sources. However, aerosol extinctions in Figure 8 (and also BC concentrations in Figure 7) are similar values at 0-300 meters above the surface, likely due to mixing

within the PBL. The small difference in sulfate would be attributed to secondary production by gas-phase or aqueous-phase reactions, rather than emission height?

Lines 293-301, it would be helpful to mention the comparison of AAOD and BC profiles, because AAOD is primarily influenced by BC.

Lines 345-346, "Gas-aerosol exchange", is this gas-phase chemical reaction of SO2?

Lines 347-351, the analysis shown in Figure 9 is interesting. I think that small contribution of the below-cloud scavenging is also interesting. What scheme is used for the below-cloud scavenging in the model calculations?

Figure 9, please describe the figure caption more carefully and clarify abbreviation (AQ chem H2SO4, SO4, etc.).

Figure 12 and Figure S10, RMSE of POM in LR simulations (Figure S10b and S10d) are much greater than those in PRM simulations (Figure 12b and 12d). On the other hand, RMSE of BC and SO4 are similar levels in all simulations (Figures 12, 14, S10, S11). Could the authors explain this, if possible?

Lines 452-471, is one-year integration sufficient to extract the difference in aerosol radiative forcings, specifically aerosol-cloud interactions, between the revised and default emission treatments? The smaller difference over NA (compared to that over the ocean) does not guarantee a validity of the analysis? Could the authors answer this, if possible?

Lines 501-502, I cannot not understand this. The authors state here the importance of the spatial heterogeneity for ACI. On the other hand, they state that the overall impact of the revised emission treatment is small for the anthropogenic aerosol forcing estimates (Lines 470-471).

**Technical comments:**

Lines 94-95, typo? "Fig. 1b", is this Fig. 1a?

Line 243, SO4 emissions → SO2 emission or sulfur emissions

Line 244, sulfate emission → sulfur emission or SO2 emission

Line 258, spatial distribution of "annual mean" surface concentration resulting …

Line 345, sulfate emissions, is this SO2 emissions or sulfur emissions?

Line 390, "Prescribed sulfate aerosol emissions", is this sulfate produced by SO2?

Line 414, typo? "from 0.44 to 0.59 for BC and 0.43 to 0.51 POM" → "from 0.43 to 0.59 for BC and 0.44 to 0.51 POM"

Lines 458-463, typo, several Fig. S13 → Fig, S12

Line 466, typo, Fig. S14a, Figure S14d → Fig. S13a, Fig. S13d

---

## Author Comment (AC1)

**Response to Review # 1**

We thank the referees for their careful review and constructive comments. We made major revisions to our manuscript in response to all the review comments, including new simulations as well as updated and newly added figures and tables. Below please find our point-by-point responses to referee # 1 (in blue).

**Referee # 1**

In this article, the authors update the pre processing and in particular the remapping of emission datasets from the original resolution to the model resolution. This is a problem faced by all atmospheric composition systems and as such, there is a possible interest from the community in such aspects. The paper is well written and organized and the plots are nicely done.

We are pleased to hear that the paper has been well-received and thankful that the referee recognizes the importance of the remapping process of emission datasets for atmospheric composition.

However, I feel that the comparison made in the paper is unfair: it is not a surprise that gradients are better represented with emissions at a resolution corresponding to ~42km grid than with emissions at 1.9x2.5° resolution. It is very possible that I misunderstood something, but why didn't the authors use the original CEDS or CMIP6 emissions at 0.25x0.25° or 0.5x0.5° as an input to the original emissions treatment (left side of Figure 2b) instead of the 1.9x2.5° resolution?

We appreciate the referee's feedback, which points out the need to provide further clarity in our model's description, particularly in explaining the choice of "default" low-resolution (1.9x2.5°) prescribed emissions. In response, we provide a more detailed discussion below on the model's standard configuration, the rationale behind using low-resolution emission data, as well as the basis for our comparisons between high-resolution and low-resolution emissions.

We want to start by emphasizing that the term "emission treatment" in our study refers to the combination of both the (1) prescribed emission input data, and (2) model routines for reading/interpolating them onto the model-native grid. Since the CMIP6 emissions are not on the model's native grids, EAM requires spatial interpolation or remapping (Figure R1). This is the source of the "interpolation error". For fluxes (i.e., emission flux), this remapping should be done conservatively (Jones 1999). The default linear remapping in the standard EAM is non-conservative and may lead to a large interpolation error (error from non-conservation in addition to the interpolation error due to resolution differences) in the standard EAM.

[Figure]

**Figure R1:** A schematic mesh representation of emission input data on regular latitude-longitude grids and EAM model-native spectral element grids. The top panels are for the EAM globally uniform (in resolution) grids, and the bottom panels are for the EAM non-uniform or RRM grids. The horizontal orange arrows represent the interpolation method used in the "default" emission treatment to remap latitude-longitude emissions to model-native emissions.

For global high-res (HR, 0.5-degree) applications (with the uniform grid as in Figure R1 top panels), we can use emission data at higher resolution, where the interpolation error is much smaller. However, linear interpolation of spatially discontinuous variables from a finer grid to a coarser model often leads to significant conservation error. For low-res (LR) and non-uniform RRM grids (Figure R1 bottom panels), directly using the high-res emissions with the linear interpolation will also lead to large conservation errors. Figures R2 and R3 illustrate the errors associated with using low- versus high-resolution emissions on coarser grids. When mass flux is not conserved, errors are exacerbated with the incorporation of high-res (0.5-degree) emissions compared to the low-res (~2-degree) emissions (Fig. R2a, b and R3a, b). These errors may propagate in model simulations and affect simulated aerosol concentrations. Figures R11 and R12 show that the incorporation of 0.5-degree emissions leads to larger differences in the simulated aerosol burden compared to the simulation with the ~2-degree emissions (simulation details in Table R1). To mitigate this issue, E3SM/EAM uses ~2-degree (1.9x2.5) emission data for the LR and RRM simulations as a part of the "default" emission treatment. For global uniform HR simulations, we often use the 0.5-degree emission data.

[Figure]

**Figure R2:** Spatial distributions of surface Black Carbon (BC) emission differences among different remapping configurations. Three remapping configurations are exploited, including the conservative remapping of the high-resolution (0.5-degree) emission data onto the EAM ~4-degree physical grids (named "0.5-deg+CON"), the non-conservative linear remapping of the 0.5-degree emission data onto the ~4-degree grids (named "0.5-deg+LIN"), and the non-conservative linear remapping of the low-resolution (2-degree) emission data onto the ~4-degree grids (named "2-deg+LIN"). The first row (a, b) shows the differences between the "2-deg+LIN" and "0.5-deg+CON" remapping configurations, the second row (c, d) is for the differences between the "0.5-deg+LIN" and "0.5-deg+CON" remapping configurations, and the third row (e, f) compares the "0.5-deg+LIN" and "2-deg+LIN" configurations. The emission differences (a, c, e) are shown in the left panels in molecules/cm$^2$/s and the relative differences (b, d, f) are shown in the right panels in percent (%).

[Figure]

**Figure R3:** Same as Figure R2 but for elevated SO₂ emissions (i.e., energy, industrial, biomass burning, and volcanic sources).

On the other hand, as the referee pointed out, using emission data on a coarse grid will lead to large "heterogeneity" errors due to loss of spatial heterogeneity of high-resolution emissions. Therefore, we propose a "revised" emission treatment in the study, which is a combination of (1) emission data at the highest available resolution and (2) EAM routines to directly read conservatively remapped data in the model-native grid. Thus, we can estimate the error caused by the "default" treatment and provide information to model users on how large errors can be induced in the simulated aerosol properties (e.g., concentrations, optical depth) and aerosol forcing. We think such estimates will be useful for users of the E3SM model or other models with similar emission treatments.

We also believe this evaluation is useful to examine whether using emission data at higher resolutions can significantly change the aerosol simulation. If this is true for 0.25-degree or 0.5-degree simulations, we should consider using even higher-resolution emission data (e.g., the 10-km CEDS emission inventory) for the cloud-permitting scale (e.g., 3 km) model simulations. For instance, the standard configuration of E3SM requires pre-processed emissions data from

CEDS and GFED. The finest resolution emissions currently available for E3SM is approximately 0.5 degrees. Meanwhile, CEDS offers an emission inventory at 10-km resolution (McDuffie, et al. 2020). Based on our findings, there is a compelling case to be made for using higher-resolution emission data to enhance the fidelity of cloud-permitting scale aerosol simulations using our revised emission treatment.

We have included the above discussion in the revised manuscript. Additionally, we have added Figures R2 and R3 as supplementary figures.

Comparing simulations with the old emissions treatment and this high resolution input with simulations with the new emissions treatment using the same input would be more meaningful, and more interesting for the reader. As such, I would recommend a major revision, and suggest to the authors that they rewrite their manuscript in order to show the added value of their new approach but using the same emission datasets as input for the two simulations. Otherwise it is hard to discriminate between the added value of the new emissions treatment and that of using higher resolution emissions (which is well known).

We agree with the assessment from the referee and have undertaken substantial revisions to address the major comments. We agree that it is useful to perform additional simulations to show the impact of the revised emission treatment. To address this, we have performed 4 additional simulations using the same high-res (~0.5-degree) emission data on the latitude-longitude grid as input. Table 1 is updated accordingly as below (Table R1).

**Table R1:** List of simulations performed and analyzed in this study. All simulations, including three low-resolution (LR, ne30pg2) simulations and three regionally refined model (RRM) simulations, are nudged toward the ERA5 reanalysis. The LR simulations have a dynamics grid spacing of ~110 km (~1 degree), while the RRM simulations have high-resolution meshes (dynamics grid spacing of ~28 km) over North America but low-resolution meshes (same as LR) for other areas. EHR indicates that high-resolution emission data (~0.5 degrees), instead of the default low-resolution data (~2 degrees), are used as input. RLL refers to the regular latitude/longitude grids. SE refers to the new emission treatment based on model native spectral element grids. Present-day (PD) and pre-industrial (PI) simulations are conducted with anthropogenic aerosol emissions from the years 2014 and 1850, respectively.

| Group | Simulation name | Model Resolution | Resolution of emission data | Remapping method |
|---|---|---|---|---|
| 1 | LR-PD (PI) | ne30pg2 | ~2 RLL | Linear interpolation |
|  | LR-EHR-PD (PI) | ne30pg2 | ~0.5 RLL | Linear interpolation |
|  | LR-SE-PD (PI) | ne30pg2 | ne30pg2 | Conservative remapping |
| 2 | RRM-PD (PI) | NA RRM | ~2 RLL | Linear interpolation |
|  | RRM-EHR-PD (PI) | NA RRM | ~0.5 RLL | Linear interpolation |
|  | RRM-SE-PD (PI) | NA RRM | NA RRM | Conservative remapping |

It's noteworthy that the new "EHR" simulations utilize the same high-resolution emissions as the "SE" simulations. Consequently, the primary differences in error estimates between these two simulations are attributed to interpolation errors and/or conservation errors. Conversely, the error estimates from the RRM-PD and LR-PD simulations include both heterogeneity and interpolation/conservation errors. Therefore, the two types of errors are not distinctly separated. However, it is possible to make an intuitive estimation of the "heterogeneity" errors alone by comparing the discrepancy in error values between the two sets of simulations. For instance, Table R2 depicts the revised Table 2 from the manuscript, which indicates that the normalized RMSE is primarily driven by the "heterogeneity" errors in RRM-PD. This deduction is reasonable given that the conservation error across North American land is comparatively small when comparing the RRM-EHR-PD and RRM-SE-PD simulations.

**Table R2:** EAMv2 anthropogenic aerosol emission data statistics in the default emission treatment for present-day (PD) RRM simulations. Statistics are shown for both the surface and elevated emissions of different aerosol species. All estimates are over the North American land (bounded by 15° – 75°N and 50°W – 170°W). Mean values indicate the area-weighted mean emission fluxes. NMB, NStdDevB, and N_RMSE are defined as $\frac{\sum(emis_{lin} - emis_{accurate})}{\sum emis_{accurate}} \times 100\%$, $\frac{stdDev_{lin} - stdDev_{accurate}}{stdDev_{accurate}}$, $\frac{RMSE}{stdDev_{accurate}} \times 100\%$ , respectively. The subscript "accurate" indicates data that preserve spatial heterogeneity and conserve mass. The subscript "lin" indicates linearly interpolated data used in the default treatment. NMB, NStdDevB, RMSE, and N_RMSE before (after) the slash are estimates for RRM-PD (RRM-EHR-PD). Units of Mean, StdDev, and RMSE are in kg m$^{-2}$s$^{-1}$. N_RMSE and NMB are in percentage (%). NStdDevB is unitless.

| Aerosol | Emission space | Mean [x 10$^{-12}$] | NMB | StdDev [x 10$^{-12}$] (accurate) | NStdDevB | RMSE [x 10$^{-12}$] | N_RMSE [%] |
|---|---|---|---|---|---|---|---|
| BC | surface | 5.52 | -8.961/ -0.27 | 12.9 | -0.395/ -0.028 | 8.71/ 1.27 | 67.4/ 9.8 |
| | elevated | 1.76 | -2.704/ 0.271 | 17.6 | -0.423/ 0.0065 | 12.7/ 1.38 | 72.2/ 7.8 |
| POM | surface | 19.8 | -10.504/ -0.341 | 51.3 | -0.369/ -0.02 | 32/ 4.18 | 62.3/ 8.3 |
| | elevated | 45 | -1.295/ 0.326 | 505 | -0.422/ 0.0078 | 363/ 38.9 | 71.7/ 7.7 |
| SO4 | surface | 0.59 | 1.025/ -0.026 | 1.24 | -0.251/ -0.015 | 0.67/ 0.0084 | 54.1/ 6.8 |
| | elevated | 5.37 | -5.039/ -1.31 | 19.1 | -0.525/ -0.057 | 16.2/ 2.71 | 84.8/ 14.1 |

To incorporate the analysis from the new "EHR" simulations in our revised manuscript, we have revised Figures 3, 12, 13, and 14 shown as Figures R4 through R7 here. Figure R4 illustrates regions on a high-resolution mesh within the North American RRM (NA RRM). In these regions, the predominant cause of inaccuracies is driven by loss of "heterogeneity" (Table R2). As a result, the data displayed in panels c and g correspond to similar patterns to those in panels d and h (revised emission treatment), while RRM-PD shows a large difference in the pattern from the other cases.

[Figure]

**Figure R4:** Spatial distribution of the present-day surface BC emissions (top) and column-integrated $SO_2$ emissions (bottom) from the original high-resolution data (a, e), the RRM-PD (default emission treatment) (b, f), the RRM-EHR-PD (c, g) and the RRM-SE-PD (revised emission treatment) (d, h) simulations. $SO_2$ emissions are taken from elevated sources (i.e., energy, industrial, biomass burning, and volcanic sources). Distributions are shown over the eastern (top row)) and western (bottom row) United States for BC ($SO_2$) emissions in kg/m$^2$/s units. Red circles in panels b and f indicate major cities with large anthropogenic BC and $SO_2$ emissions respectively. Markers titled 1, 2, 3, 4, 5, 6, and 7 depict Boston, New York, Chicago, Toronto, Montreal, Los Angeles, and San Francisco respectively.

[Figure]

**Figure R5:** Scatter plots between simulated and observed monthly mean surface concentrations of (a, c) Black Carbon (BC) and (b, d) Primary Organic Matter (POM). Observations of the surface concentrations are from IMPROVE for the simulation year of 2016. Scatter plot statistics compare the Spearman's correlation (R), number of data points (n), RMSE, NMB values between (a, b) RRM-PD, (c, d) RRM-EHR-PD, and (e, f) RRM-SE-PD simulation. RMSE and NMB are defined in the caption of Table 2. Solid lines indicate the 1:1 ratio, and the dashed lines indicate the 1:2 and 2:1 ratio. The values at the top of each column indicate the observed mean.

[Figure]

[Figure]

**Figure R6:** Boxplot comparison of the daily mean distribution for (a) BC and (b) POM surface concentrations from RRM-PD simulation, RRM-SE-PD simulation, RRM-EHR-PD simulation and IMPROVE network measurements. The whiskers are based on 1.5 times interquartile range (IQR). Distributions are plotted for different seasons over the simulation year, with red diamonds indicating the seasonal means.

Since over North America the high-resolution emission data are used and the heterogeneity error is small for RRM-EHR-PD, as expected, we don't see substantial differences between RRM-SE-PD and RRM-EHR-PD simulations (Figures R5, R6, and R7). In contrast, we found significant errors over coarse grid regions, which are described later to explain the influence of mass conservation and/or interpolation errors.

We made additional plots of the simulated spatial distributions as in Figures 7 and 8 to illustrate the difference between RRM-SE-PD and RRM-EHR-PD (Figures R8 and R9). These are included as supplementary figures in the revised manuscript. Figures R7 and R8 show reduced normalized RMSE (<= 10%) from RRM-EHR-PD (compared to RRM-PD) in simulating aerosol surface concentrations and optical properties over North America. As described above, these results are expected since most of the errors over high-res RRM mesh are driven by the "heterogeneity" errors in RRM-PD (Table R2). In addition, we have added a supplementary figure to illustrate larger discrepancies in high-frequency aerosol concentration profiles. Figure R10 shows differences in simulated daily mean BC, POM, and Sulfate concentration profiles and column-integrated burden between RRM-SE-PD and RRM-EHR-PD. We found significant errors (>10%) can exist from loss of conservation alone in high-frequency data.

[Figure]

**Figure R7:** Scatter plots between simulated and observed monthly mean surface concentrations of (a, c) sulfate (SO₄) aerosols and (b, d) Aerosol Optical Depth (AOD) at 550 nm. Observations of the surface concentrations and AOD are from IMPROVE and AERONET respectively for the simulation year of 2016. Scatter plot statistics include the Spearman's correlation (R), number of data points (n), RMSE, NMB values between (a, b) RRM-PD, (c, d) RRM-SE-PD, and (e, f) RRM-EHR-PD simulation. RMSE and NMB are defined in Table 2. Solid lines indicate the 1:1 ratio, and the dashed lines indicate the 1:2 and 2:1 ratio. The values at the top of each column indicate the observed mean.

[Figure]

**Figure R8:** Simulated spatial distribution of annual mean aerosol surface concentration from RRM-EHR-PD (left column) and the relative difference between RRM-EHR-PD and RRM-SE-PD (right column) over North America. Distributions are shown for (a, b) Black Carbon (BC), (c, d) Primary Organic Matter (POM), and (e, f) Sulfate aerosols. The relative difference for field X is calculated as: $\left(\frac{X_{ehr}-X_{se}}{X_{se}}\right) \times 100\%$, where "se" and "ehr" subscripts refer to the simulations with revised and RRM-EHR-PD emission treatment, respectively. Mean, RMSE, and normalized RMSE (N_RMSE) are indicated at the top right corner of the panels. Mean and RMSE have units of μg m$^{-3}$. N_RMSE is defined in Table 2.

[Figure]

**Figure R9:** Spatial distribution of annual mean simulated (a, b) aerosol extinction at the surface, (c, d) aerosol absorption at the surface, (e, f) Aerosol Optical Depth (AOD), and (g, h) absorbing AOD from RRM-EHR-PD (left column) and the relative difference between RRM-EHR-PD and RRM-SE-PD (right column) over North America. The relative difference for field X is calculated as: $\left(\frac{X_{ehr}-X_{se}}{X_{se}}\right) \times 100\%$, where "se" and "ehr" subscripts refer to the simulations with revised and RRM-EHR-PD emission treatment respectively. Mean, RMSE, and normalized RMSE (N_RMSE) are indicated at the top right corner of the panels. Mean and RMSE have units of μg m$^{-3}$. N_RMSE is defined as in Table 2.

[Figure]

**Figure R10:** Daily mean concentration profile and burden time-series of (a-d) BC, (e-h) POM, and (i-l) Sulfate aerosols. All profiles are shown during the month of July 2016 over highly polluted areas in eastern North America. Simulated vertical distribution and burden time series from RRM-EHR-PD (left column) and the relative difference between RRM-EHR-PD and RRM-SE-PD (right column) are shown.

A side topic of the paper could be the mass conservative aspect of their revised treatment of aerosol emissions – how much does it change emissions as compared to the (I suppose) non mass conservative remapping/interpolation used before? With what impacts on simulated aerosol burden/surface concentration? I think also more detail should be given as to how/with which method is mass conserved in the revised treatment.

We agree that the impact of mass conservation should be described in the manuscript more explicitly. We want to emphasize that the "revised" emission treatment has no conservation error. Since there is little to no heterogeneity difference between RRM-EHR-PD and RRM-SE-PD (Fig. R4), we can compare them to estimate the impact of conservation errors. Therefore, Table R2 and Figure R5-R10 show the emission and simulation errors over finer grids, when emission mass is not conserved. On the other hand, we expect larger impacts over coarser grids (Table R3). For instance, Figures R2c, d and R3c, d illustrate the mass conservation errors in BC and $SO_2$ emissions over coarser grids.

**Table R3:** EAMv2 anthropogenic aerosol emissions data statistics in the default emission treatment for present-day (PD) RRM simulations over coarser grids. All estimates are over the South Asian land surface (bounded by $0° - 30°$ N and $60°$ E $- 120°$ E). Statistics are shown for both the surface and elevated emissions of different aerosol species. Mean values indicate the area-weighted mean emission fluxes. NMB, NStdDevB, and N_RMSE are defined as $\frac{\sum(emis_{lin} - emis_{accurate})}{\sum emis_{accurate}} \times 100\%, \frac{stdDev_{lin} - stdDev_{accurate}}{stdDev_{accurate}}, \frac{RMSE}{stdDev_{accurate}} \times 100\%$, respectively. The "accurate" subscript indicates data that preserve spatial heterogeneity and conserve mass. The "lin" subscript indicates linearly interpolated data used in the default treatment. NMB, NStdDevB, RMSE, and N_RMSE estimates are from RRM-PD/RRM-EHR-PD emission data. Units of Mean, StdDev, and RMSE are in $kg/m^2/s$. N_RMSE and NMB are in percentage (%). NStdDevB is unitless.

| Aerosol | Emission space | Mean [x $10^{-12}$] | NMB | StdDev [x $10^{-12}$] (accurate) | NStdDevB | RMSE [x $10^{-12}$] | N_RMSE [%] |
|---------|---------------|---------------------|-----|----------------------------------|----------|---------------------|------------|
| BC | surface | 76.3 | -3.091/ 0.734 | 84.6 | -0.236/ 0.192 | 42.6/ 30.3 | 50.3/ 35.7 |
|  | elevated | 4.1 | -2.566/ 4.997 | 9.49 | -0.244/ 0.340 | 4.48/ 4.89 | 47.1/ 51.4 |
| POM | surface | 286 | -3.088/ 0.769 | 278 | -0.202/ 0.151 | 120/ 85.8 | 43.3/ 30.9 |
|  | elevated | 59.6 | -2.414/ 2.032 | 177 | -0.285/ 0.306 | 94.4/ 86 | 53.4/ 48.6 |
| SO4 | surface | 5.56 | -2.927/ 0.671 | 7.38 | -0.223/ 0.168 | 3.62/ 2.27 | 49/ 30.7 |
|  | elevated | 32.4 | -4.259/ 0.023 | 64.8 | -0.381/ 0.598 | 43.8/ 57.5 | 67.6/ 88.7 |

Figures R11 and R12 show the simulated BC and sulfate burden differences between RRM-PD, RRM-EHR-PD, and RRM-SE-PD. In the refined region (NA), the overall differences between the simulations are small. While in other regions (e.g., East Asia), the differences are much larger, where employing the default treatment (RRM-PD, 2deg emission) results in smaller burden differences from RRM-SE-PD (R11b and R12b) compared to from RRM-EHR-PD that uses the high-resolution emission (R11d and R12d).

[Figure]

**Figure R11:** Global distribution of simulated BC aerosol burden differences between (a, b) RRM-PD and RRM-SE-PD, (c, d) RRM-EHR-PD and RRM-SE-PD, and (e, f) RRM-EHR-PD and RRM-PD simulations. All column-integrated burden absolute differences are shown in μg m$^{-2}$ and the relative differences are shown in percent (%).

[Figure]

**Figure R12:** Global distribution of simulated sulfate aerosol burden differences between (a, b) RRM-PD and RRM-SE-PD, (c, d) RRM-EHR-PD and RRM-SE-PD, and (e, f) RRM-EHR-PD and RRM-PD simulations. All column-integrated burden absolute differences are shown in μgm$^{-2}$ and the relative differences are shown in percent (%).

We have included the above discussion along with Figures R11 and R12 in the revised manuscript to describe the impact of non-conservative remapping. We have also added Figures R2, R3, and Table R3 as part of the supplementary materials.

**Works Cited**

Jones, Philip W. 1999. "First-and second-order conservative remapping schemes for grids in spherical coordinates." *Monthly Weather Review* 2204-2210.

McDuffie, Erin E, Steven J Smith, Patrick O'Rourke, Kushal Tibrewal, Ch Venkataraman, ra, Eloise A Marais, et al. 2020. "A global anthropogenic emission inventory of atmospheric pollutants from sector-and fuel-specific sources (1970--2017): an application of the Community Emissions Data System (CEDS)." *Earth System Science Data* 3413-3442.

---

## Author Comment (AC2)

**Response to Review # 2**

We thank the referees for their careful review and constructive comments. We made major revisions to our manuscript in response to all the review comments, including new simulations as well as updated and newly added figures and tables. Below please find our point-by-point responses to referee # 2 (in blue).

**General comments:**

This work implements a revised emission treatment in E3SM to preserve the original emission spatial heterogeneity and conserve emission mass fluxes in the simulations. The authors show significant differences in the simulated surface concentration of aerosols between the default and revised emission treatments in regionally-refined high-resolution simulations. They also show that the revised emission treatment leads to improved heterogeneity in simulated surface concentration of aerosols, particularly in regions with sharp emission gradients.

This study is interesting and the subject is of great interest to GMD. The manuscript is well written. However, the interpretations of the simulations with low-resolution emissions and the comparisons are not satisfactory, which are cause for concern (see Major comments). Most other comments listed below are minor clarifications. Once these points are addressed satisfactory, the paper should in my opinion be suitable for publication in GMD.

We appreciate the referee's positive feedback and have undertaken substantial revisions to address the major comments. Additionally, we have made changes for further clarification based on the minor and technical comments. Our detailed responses are presented below.

**Major comments:**

I cannot understand the author's intention about the simulations and comparisons. I may misunderstand something, but I give comments here.

I think that it is common to use the original emission data for model simulations. I cannot understand why the authors do not use the original CMIP6 emissions (CEDS 0.5° x 0.5° and GFED4 0.25° x 0.25°) as inputs to the default emission treatments (LR-PD), instead of the default low-resolution (1.9° x 2.5°) prescribed emissions. Is this a problem specific to the model used in this study?

I cannot understand why the authors conduct high-resolution (42 km) simulation (RRM-PD) with much low-resolution emission (1.9° x 2.5°). Because there is a large difference in the resolutions between two (42 km and 1.9° x 2.5°), it would be qualitatively obvious that high-resolution simulations with low-resolution emission cannot reproduce aerosol concentrations at the surface in highly polluted regions. Even though the authors understand these things, do they have some other purpose in performing this simulation, such as an evaluation of the impacts in advance before doing cloud-resolving scale simulations with relatively low-resolution (0.25°-0.5°) emissions (although the evaluation would be difficult)?

We appreciate the referee's feedback, which points out the need to provide further clarity in our model's description, particularly in explaining the choice of "default" low-resolution (1.9x2.5°) prescribed emissions. In response, we provide a more detailed discussion below on the model's standard configuration, the rationale behind using low-resolution emission data, as well as the basis for our comparisons between high-resolution and low-resolution emissions.

We want to start by emphasizing that the term "emission treatment" in our study refers to the combination of both the (1) prescribed emission input data, and (2) model routines for reading/interpolating them onto the model-native grid. Since the CMIP6 emissions are not on the model's native grids, EAM requires spatial interpolation or remapping (Figure R1). This is the source of the "interpolation error". For fluxes (i.e., emission flux), this remapping should be done conservatively (Jones 1999). The default linear remapping in the standard EAM is non-conservative and may lead to a large interpolation error (error from non-conservation in addition to the interpolation error due to resolution differences) in the standard EAM.

[Figure]

**Figure R1:** A schematic mesh representation of emission input data on regular latitude-longitude grids and EAM model-native spectral element grids. The top panels are for the EAM globally uniform (in resolution) grids, and the bottom panels are for the EAM non-uniform or RRM grids. The horizontal orange arrows represent the interpolation method used in the "default" emission treatment to remap latitude-longitude emissions to model-native emissions.

For global high-res (HR, 0.5-degree) applications (with the uniform grid as in Figure R1 top panels), we can use emission data at higher resolution, where the interpolation error is much smaller. However, linear interpolation of spatially discontinuous variables from a finer grid to a

coarser model often leads to significant conservation error. For low-res (LR) and non-uniform RRM grids (Figure R1 bottom panels), directly using the high-res emissions with the linear interpolation will also lead to large conservation errors. Figures R2 and R3 illustrate the errors associated with using low- versus high-resolution emissions on coarser grids. When mass flux is not conserved, errors are exacerbated with the incorporation of high-res (0.5-degree) emissions compared to the low-res (~2-degree) emissions (Fig. R2a, b and R3a, b). These errors may propagate in model simulations and affect simulated aerosol concentrations. Figures R11 and R12 show that the incorporation of 0.5-degree emissions leads to larger differences in the simulated aerosol burden compared to the simulation with the ~2-degree emissions (simulation details in Table R1). To mitigate this issue, E3SM/EAM uses ~2-degree (1.9x2.5) emission data for the LR and RRM simulations as a part of the "default" emission treatment. For global uniform HR simulations, we often use the 0.5-degree emission data.

[Figure]

**Figure R2:** Spatial distributions of surface Black Carbon (BC) emission differences among different remapping configurations. Three remapping configurations are exploited, including

the conservative remapping of the high-resolution (0.5-degree) emission data onto the EAM ~4-degree physical grids (named "0.5-deg+CON"), the non-conservative linear remapping of the 0.5-degree emission data onto the ~4-degree grids (named "0.5-deg+LIN"), and the non-conservative linear remapping of the low-resolution (2-degree) emission data onto the ~4-degree grids (named "2-deg+LIN"). The first row (a, b) shows the differences between the "2-deg+LIN" and "0.5-deg+CON" remapping configurations, the second row (c, d) is for the differences between the "0.5-deg+LIN" and "0.5-deg+CON" remapping configurations, and the third row (e, f) compares the "0.5-deg+LIN" and "2-deg+LIN" configurations. The emission differences (a, c, e) are shown in the left panels in molecules/cm$^2$/s and the relative differences (b, d, f) are shown in the right panels in percent (%).

[Figure]

**Figure R3:** Same as Figure R2 but for elevated SO$_2$ emissions (i.e., energy, industrial, biomass burning, and volcanic sources).

On the other hand, as the referee pointed out, using emission data on a coarse grid will lead to large "heterogeneity" errors due to loss of spatial heterogeneity of high-resolution emissions. Therefore, we propose a "revised" emission treatment in the study, which is a combination of (1) emission data at the highest available resolution and (2) EAM routines to directly read conservatively remapped data in the model-native grid. Thus, we can estimate the error caused by the "default" treatment and provide information to model users on how large errors can be induced in the simulated aerosol properties (e.g., concentrations, optical depth) and aerosol forcing. We think such estimates will be useful for users of the E3SM model or other models with similar emission treatments.

We also believe this evaluation is useful to examine whether using emission data at higher resolutions can significantly change the aerosol simulation. If this is true for 0.25-degree or 0.5-degree simulations, we should consider using even higher-resolution emission data (e.g., the 10-km CEDS emission inventory) for the cloud-permitting scale (e.g., 3 km) model simulations. For instance, the standard configuration of E3SM requires pre-processed emissions data from CEDS and GFED. The finest resolution emissions currently available for E3SM is approximately 0.5 degrees. Meanwhile, CEDS offers an emission inventory at 10-km resolution (McDuffie, et al. 2020). Based on our findings, there is a compelling case to be made for using higher-resolution emission data to enhance the fidelity of cloud-permitting scale aerosol simulations using our revised emission treatment.

We have included the above discussion in the revised manuscript. Additionally, we have added Figures R2 and R3 as supplementary figures.

Although the authors basically compare RRM-PD and RRM-SE-PD simulations or compare LR-PD and LR-SE-PD simulations, comparison between LR-SE-PD and RRM-SE-PD simulations (using the same original CMIP6 emissions?) would be meaningful. For example, there is a better agreement between simulated and observed BC and POM in RRM-SE-PD (Figure 12c-d) than those in LR-SE-PD (Figure S10c-d).

We agree with the referee that comparing LR-SE-PD and RRM-SE-PD would be meaningful since in terms of emission treatment they both use conservative remapping and the emissions are accurate at their respective resolutions (with different heterogeneity). However, the simulation difference, e.g., the improved agreement between simulated and observed surface concentrations for species, is not only driven by emission differences but also by sensitivities related to other resolution-dependent factors, such as the cloud processes and aerosol microphysics in the host model (Li, et al. 2023). In the revised manuscript, we have clarified that our primary focus for this study is on the impact of different emission implementations wherever appropriate.

If the authors want to show influences of improvement of the heterogeneity and mass conservation separately, additional low-resolution (165 km) simulation and high-resolution (42 km) simulation using the same original CMIP6 emissions as inputs to the default emission

treatments and comparisons with other four simulations (shown in Table 1) would be helpful. However, because comparison between LR-PD and LR-SE-PD simulations does not show significant difference in surface aerosol concentrations and AOD (Lines 443-446, Figures S10-S11), it may be difficult to evaluate the quantitative separation.

We agree with the assessment from the referee and have undertaken substantial revisions to address the major comments. We agree that it is useful to perform additional simulations to show the impact of the revised emission treatment. To address this, we have performed 4 additional simulations using the same high-res (~0.5-degree) emission data on the latitude-longitude grid as input. Table 1 is updated accordingly as below (Table R1).

**Table R1:** List of simulations performed and analyzed in this study. All simulations, including three low-resolution (LR, ne30pg2) simulations and three regionally refined model (RRM) simulations, are nudged toward the ERA5 reanalysis. The LR simulations have a dynamics grid spacing of ~110 km (~1 degree), while the RRM simulations have high-resolution meshes (dynamics grid spacing of ~28 km) over North America but low-resolution meshes (same as LR) for other areas. EHR indicates that high-resolution emission data (~0.5 degrees), instead of the default low-resolution data (~2 degrees), are used as input. RLL refers to the regular latitude/longitude grids. SE refers to the new emission treatment based on model native spectral element grids. Present-day (PD) and pre-industrial (PI) simulations are conducted with anthropogenic aerosol emissions from the years 2014 and 1850, respectively.

| Group | Simulation name | Model Resolution | Resolution of emission data | Remapping method |
|-------|-----------------|------------------|------------------------------|------------------|
| 1 | LR-PD (PI) | ne30pg2 | ~2 RLL | Linear interpolation |
| | LR-EHR-PD (PI) | ne30pg2 | ~0.5 RLL | Linear interpolation |
| | LR-SE-PD (PI) | ne30pg2 | ne30pg2 | Conservative remapping |
| 2 | RRM-PD (PI) | NA RRM | ~2 RLL | Linear interpolation |
| | RRM-EHR-PD (PI) | NA RRM | ~0.5 RLL | Linear interpolation |
| | RRM-SE-PD (PI) | NA RRM | NA RRM | Conservative remapping |

It's noteworthy that the new "EHR" simulations utilize the same high-resolution emissions as the "SE" simulations. Consequently, the primary differences in error estimates between these two simulations are attributed to interpolation errors and/or conservation errors. Conversely, the error estimates from the RRM-PD and LR-PD simulations include both heterogeneity and interpolation/conservation errors. Therefore, the two types of errors are not distinctly separated. However, it is possible to make an intuitive estimation of the "heterogeneity" errors alone by comparing the discrepancy in error values between the two sets of simulations. For instance, Table R2 depicts the revised Table 2 from the manuscript, which indicates that the normalized RMSE is primarily driven by the "heterogeneity" errors in RRM-PD. This deduction is

reasonable given that the conservation error across North American land is comparatively small when comparing the RRM-EHR-PD and RRM-SE-PD simulations.

**Table R2:** EAMv2 anthropogenic aerosol emissions data statistics in the default emission treatment for present-day (PD) RRM simulations. Statistics are shown for both the surface and elevated emissions of different aerosol species. All estimates are over the North American land (bounded by 15° – 75°N and 50°W – 170°W). Mean values indicate the area-weighted mean emission fluxes. NMB, NStdDevB, and N_RMSE are defined as $\frac{\sum (emis_{lin} - emis_{accurate})}{\sum emis_{accurate}} \times 100\%$, $\frac{stdDev_{lin} - stdDev_{accurate}}{stdDev_{accurate}}$, $\frac{RMSE}{stdDev_{accurate}} \times 100\%$ respectively. The subscript "accurate" indicates data that preserve spatial heterogeneity and conserve mass. The subscript "lin" indicates linearly interpolated data used in the default treatment. NMB, NStdDevB, RMSE, and N_RMSE before (after) the slash are estimates for RRM-PD (RRM-EHR-PD). Units of Mean, StdDev, and RMSE are in kg m$^{-2}$s$^{-1}$. N_RMSE and NMB are in percentage (%). NStdDevB is unitless.

| Aerosol | Emission space | Mean [x 10$^{-12}$] | NMB | StdDev [x 10$^{-12}$] (accurate) | NStdDevB | RMSE [x 10$^{-12}$] | N_RMSE [%] |
|---|---|---|---|---|---|---|---|
| BC | surface | 5.52 | -8.961/ -0.27 | 12.9 | -0.395/ -0.028 | 8.71/ 1.27 | 67.4/ 9.8 |
| | elevated | 1.76 | -2.704/ 0.271 | 17.6 | -0.423/ 0.0065 | 12.7/ 1.38 | 72.2/ 7.8 |
| POM | surface | 19.8 | -10.504/ -0.341 | 51.3 | -0.369/ -0.02 | 32/ 4.18 | 62.3/ 8.3 |
| | elevated | 45 | -1.295/ 0.326 | 505 | -0.422/ 0.0078 | 363/ 38.9 | 71.7/ 7.7 |
| SO4 | surface | 0.59 | 1.025/ -0.026 | 1.24 | -0.251/ -0.015 | 0.67/ 0.0084 | 54.1/ 6.8 |
| | elevated | 5.37 | -5.039/ -1.31 | 19.1 | -0.525/ -0.057 | 16.2/ 2.71 | 84.8/ 14.1 |

To incorporate the analysis from the new "EHR" simulations in our revised manuscript, we have revised Figures 3, 12, 13, and 14 shown as Figures R4 through R7 here. Figure R4 illustrates regions on a high-resolution mesh within the North American RRM (NA RRM). In these regions, the predominant cause of inaccuracies is driven by loss of "heterogeneity" (Table R2). As a result, the data displayed in panels c and g correspond to similar patterns to those in panels d and h (revised emission treatment), while RRM-PD shows a large difference in the pattern from the other cases.

[Figure]

**Figure R4:** Spatial distribution of the present-day surface BC emissions (top) and column-integrated $SO_2$ emissions (bottom) from the original high-resolution data (a, e), the RRM-PD (default emission treatment) (b, f), the RRM-EHR-PD (c, g) and the RRM-SE-PD (revised emission treatment) (d, h) simulations. $SO_2$ emissions are taken from elevated sources (i.e., energy, industrial, biomass burning, and volcanic sources). Distributions are shown over the eastern (top row)) and western (bottom row) United States for BC ($SO_2$) emissions in kg/m$^2$/s units. Red circles in panels b and f indicate major cities with large anthropogenic BC and $SO_2$ emissions respectively. Markers titled 1, 2, 3, 4, 5, 6, and 7 depict Boston, New York, Chicago, Toronto, Montreal, Los Angeles, and San Francisco respectively.

[Figure]

**Figure R5:** Scatter plots between simulated and observed monthly mean surface concentrations of (a, c) Black Carbon (BC) and (b, d) Primary Organic Matter (POM). Observations of the surface concentrations are from IMPROVE for the simulation year of 2016. Scatter plot statistics compare the Spearman's correlation (R), number of data points (n), RMSE, NMB values between (a, b) RRM-PD, (c, d) RRM-EHR-PD, and (e, f) RRM-SE-PD simulation. RMSE and NMB are defined in the caption of Table 2. Solid lines indicate the 1:1 ratio, and the dashed lines indicate the 1:2 and 2:1 ratio. The values at the top of each column indicate the observed mean.

[Figure]

[Figure]

**Figure R6:** Boxplot comparison of the daily mean distribution for (a) BC and (b) POM surface concentrations from RRM-PD simulation, RRM-SE-PD simulation, RRM-EHR-PD simulation and IMPROVE network measurements. The whiskers are based on 1.5 times interquartile range (IQR). Distributions are plotted for different seasons over the simulation year, with red diamonds indicating the seasonal means.

Since over North America the high-resolution emission data are used and the heterogeneity error is small for RRM-EHR-PD, as expected, we don't see substantial differences between RRM-SE-PD and RRM-EHR-PD simulations (Figures R5, R6, and R7). In contrast, we found significant errors over coarse grid regions, which are described later to explain the influence of mass conservation and/or interpolation errors.

We made additional plots of the simulated spatial distributions as in Figures 7 and 8 to illustrate the difference between RRM-SE-PD and RRM-EHR-PD (Figures R8 and R9). These are included as supplementary figures in the revised manuscript. Figures R7 and R8 show reduced normalized RMSE (<= 10%) from RRM-EHR-PD (compared to RRM-PD) in simulating aerosol surface concentrations and optical properties over North America. As described above, these results are expected since most of the errors over high-res RRM mesh are driven by the "heterogeneity" errors in RRM-PD (Table R2). In addition, we have added a supplementary figure to illustrate larger discrepancies in high-frequency aerosol concentration profiles. Figure R10 shows differences in simulated daily mean BC, POM, and Sulfate concentration profiles and column-integrated burden between RRM-SE-PD and RRM-EHR-PD. We found significant errors (>10%) can exist from loss of conservation alone in high-frequency data.

[Figure]

**Figure R7:** Scatter plots between simulated and observed monthly mean surface concentrations of (a, c) sulfate (SO₄) aerosols and (b, d) Aerosol Optical Depth (AOD) at 550 nm. Observations of the surface concentrations and AOD are from IMPROVE and AERONET respectively for the simulation year of 2016. Scatter plot statistics compare the Spearman's correlation (R), number of data points (n), RMSE, NMB values between (a, b) RRM-PD, (c, d) RRM-SE-PD, and (e, f) RRM-EHR-PD simulation. RMSE and NMB are defined in Table 2. Solid lines indicate the 1:1 ratio, and the dashed lines indicate the 1:2 and 2:1 ratio. The values at the top of each column indicate the observed mean.

[Figure]

**Figure R8:** Simulated spatial distribution of annual mean aerosol surface concentration from RRM-EHR-PD (left column) and the relative difference between RRM-EHR-PD and RRM-SE-PD (right column) over North America. Distributions are shown for (a, b) Black Carbon (BC), (c, d) Primary Organic Matter (POM), and (e, f) Sulfate aerosols. The relative difference for field X is calculated as: $\left(\frac{X_{ehr}-X_{se}}{X_{se}}\right) \times 100\%$, where "se" and "ehr" subscripts refer to the simulations with revised and RRM-EHR-PD emission treatment respectively. Mean, RMSE, and normalized RMSE (N_RMSE) are indicated at the top right corner of the panels. Mean and RMSE has a unit of μg m[-3]. N_RMSE is defined in Table 2.

[Figure]

**Figure R9:** Spatial distribution of annual mean simulated (a, b) aerosol extinction at the surface, (c, d) aerosol absorption at the surface, (e, f) Aerosol Optical Depth (AOD), and (g, h) absorbing AOD from RRM-EHR-PD (left column) and the relative difference between RRM-EHR-PD and RRM-SE-PD (right column) over North America. The relative difference for field X is calculated as: $\left(\frac{X_{ehr} - X_{se}}{X_{se}}\right) \times 100\%$, where "se" and "ehr" subscripts refer to the simulations with revised and RRM-EHR-PD emission treatment respectively. Mean, RMSE, and normalized RMSE (N_RMSE) are indicated at the top right corner of the panels. Mean and RMSE has a unit of µg m$^{-3}$. N_RMSE is defined as in Table 2.

[Figure]

**Figure R10:** Daily mean concentration profile and burden time-series of (a-d) BC, (e-h) POM, and (i-l) Sulfate aerosols. All profiles are shown during the month of July of 2016 over highly polluted location in eastern North America. Simulated vertical distribution and burden time-series from RRM-EHR-PD (left column) and the relative difference between RRM-EHR-PD and RRM-SE-PD (right column) are shown.

The authors should explain the objectives of the simulations and their comparisons in more details. It is unclear to me.

The revised list of simulations is shown in Table R1. We list the objectives of our simulations below:
1. We compare LR-PD with LR-SE-PD and RRM-PD with RRM-SE-PD to estimate errors from the "default" emission treatments. These errors are driven by both "heterogeneity" and conservation and/or interpolation errors. We identify how aerosol species and processes are disproportionately impacted by these treatments and how they might affect model evaluations when compared to real-world observations.
2. To identify the impact of interpolation error (that leads to conservation error) only, we compare LR-EHR-PD with LR-SE-PD as well as RRM-EHR-PD with RRM-SE-PD.
3. Comparing the error estimates from point no. 1 and 2 can help us identify the impact of the loss of emission heterogeneity in the standard model.
4. To identify the impact of the loss of emission heterogeneity in the standard model, we compare LR-PD with LR-EHR-PD and RRM-PD with RRM-EHR-PD. All simulations use the original emission treatment, but emissions at different resolutions (latitude-longitude grid) are used for LR and RRM.
5. Present-day ("PD") simulations are used for our model assessments. Meanwhile, pre-industrial ("PI") simulations are used with PD simulations to calculate aerosol radiative forcings.

The above description has been added to the revised manuscript.

The authors point out the problem of mass conservation in the default emission treatment, however, it seems that there is almost no description about them. Could the authors add explanations about influences of the mass conservation?

Thanks for the comment. We agree that the impact of mass conservation should be described in the manuscript more explicitly. We want to emphasize that the "revised" emission treatment has no conservation error. Since there is little to no heterogeneity difference between RRM-EHR-PD and RRM-SE-PD (Fig. R4), we can compare them to estimate the impact of conservation errors. Therefore, Table R2 and Figure R5-R10 show the emission and simulation errors over finer grids, when emission mass is not conserved. On the other hand, we expect larger impacts over coarser grids (Table R3). For instance, Figures R2c, d and R3c, d illustrate the mass conservation errors in BC and $SO_2$ emissions over coarser grids.

Table R3: EAMv2 anthropogenic aerosol emissions data statistics in the default emission treatment for present-day (PD) RRM simulations over coarser grids. All estimates are over the South Asian land surface (bounded by $0° - 30°$ N and $60°$ E $- 120°$ E). Statistics are shown for both the surface and elevated emissions of different aerosol species. Mean values indicate the area weighted mean emission fluxes. NMB, NStdDevB, and N_RMSE are defined as
$$\frac{\sum(emis_{lin} - emis_{accurate})}{\sum emis_{accurate}} \times 100\%, \frac{stdDev_{lin} - stdDev_{accurate}}{stdDev_{accurate}}, \frac{RMSE}{stdDev_{accurate}} \times 100\% \text{ respectively. The}$$

"accurate" subscript indicates data that preserve spatial heterogeneity and conserve mass. The "lin" subscript indicates linearly interpolated data used in the default treatment. NMB, NStdDevB, RMSE, and N_RMSE estimates are from RRM-PD/RRM-EHR-PD emission data. Units of Mean, StdDev, and RMSE are in kg/m$^2$/s. N_RMSE and NMB are in percentage (%). NStdDevB is unitless.

| Aerosol | Emission space | Mean [x 10$^{-12}$] | NMB | StdDev [x 10$^{-12}$] (accurate) | NStdDevB | RMSE [x 10$^{-12}$] | N_RMSE [%] |
|---------|---------|---------|---------|---------|---------|---------|---------|
| BC | surface | 76.3 | -3.091/ 0.734 | 84.6 | -0.236/ 0.192 | 42.6/ 30.3 | 50.3/ 35.7 |
|  | elevated | 4.1 | -2.566/ 4.997 | 9.49 | -0.244/ 0.340 | 4.48/ 4.89 | 47.1/ 51.4 |
| POM | surface | 286 | -3.088/ 0.769 | 278 | -0.202/ 0.151 | 120/ 85.8 | 43.3/ 30.9 |
|  | elevated | 59.6 | -2.414/ 2.032 | 177 | -0.285/ 0.306 | 94.4/ 86 | 53.4/ 48.6 |
| SO4 | surface | 5.56 | -2.927/ 0.671 | 7.38 | -0.223/ 0.168 | 3.62/ 2.27 | 49/ 30.7 |
|  | elevated | 32.4 | -4.259/ 0.023 | 64.8 | -0.381/ 0.598 | 43.8/ 57.5 | 67.6/ 88.7 |

Figures R11 and R12 show the simulated BC and sulfate burden differences between RRM-PD, RRM-EHR-PD, and RRM-SE-PD. In the refined region (NA), the overall differences between the simulations are small. While in other regions (e.g., East Asia), the differences are much larger, where employing the default treatment (RRM-PD, 2deg emission) results in smaller burden differences from RRM-SE-PD (R11b and R12b) compared to from RRM-EHR-PD that uses the high-resolution emission (R11d and R12d).

[Figure]

**Figure R11:** Global distribution of simulated BC aerosol burden differences between (a, b) RRM-PD and RRM-SE-PD, (c, d) RRM-EHR-PD and RRM-SE-PD, and (e, f) RRM-EHR-PD and RRM-PD simulations. All column-integrated burden absolute differences are shown in μgm$^{-2}$ and the relative differences are shown in percent (%).

[Figure]

**Figure R12:** Global distribution of simulated sulfate aerosol burden differences between (a, b) RRM-PD and RRM-SE-PD, (c, d) RRM-EHR-PD and RRM-SE-PD, and (e, f) RRM-EHR-PD and RRM-PD simulations. All column-integrated burden absolute differences are shown in μgm$^{-2}$ and the relative differences are shown in percent (%).

We included the above description with Figures R11 and R12 in the revised manuscript to describe the impact of non-conservative remapping. We also added Figures R2, R3, and Table R3 as supplementary figures and table for support.

**Specific comments:**

Lines 165-171, please describe the horizontal resolution of the reanalysis data used for the nudging.

We have added the following statement for clarification:

"The original ERA5 reanalysis data (used for preparing the nudging data) are 3-hourly data at 0.25-degree latitude-longitude resolution."

Lines 261-264 and 428-432, the authors state that the small difference in sulfate aerosols is attributed to the fact that prescribed sulfur emissions are mostly emitted from elevated sources. However, aerosol extinctions in Figure 8 (and also BC concentrations in Figure 7) are similar values at 0-300 meters above the surface, likely due to mixing within the PBL. The small difference in sulfate would be attributed to secondary production by gas-phase or aqueous-phase reactions, rather than emission height?

We agree that chemical production and vertical mixing might play a role in affecting the surface sulfate concentrations. The statement has now been revised to:

"This can be partially attributed to the fact that prescribed sulfur emissions are mostly emitted from elevated sources, such as industrial and energy sectors, as opposed to BC and POM emissions, which are primarily emitted at surface level. This difference can also be driven by mixing and chemical production within the boundary layer."

Lines 293-301, it would be helpful to mention the comparison of AAOD and BC profiles, because AAOD is primarily influenced by BC.

We thank the referee for this comment. In the original manuscript, the BC/POM profiles and the AOD/AAOD time series are sampled at two locations, where we find large differences between the simulations. To identify the relationship between absorption and BC profiles, we revised Figure 7 (a, b) so that the selected highly polluted location is the same as in Figure 8 for consistency in BC and absorption profiles. These two profiles are shown in Figure R13.

[Figure]

**Figure R13:** Daily mean BC concentration and absorption time-series during July 2016 over a highly polluted location in eastern North America (42°N and 70°W). Simulated aerosol concentration profile, absorption profile, burden, and absorption AOD (AAOD) time-series from RRM-PD (left column) and the relative difference between RRM-PD and RRM-SE-PD (right column) are shown.

Figure R13 shows that the simulated high-frequency absorption profile closely follows the changes in the BC profile, with large relative differences in the boundary layer. To highlight this relationship, we added the following statement in section 3.2.2:

"The time evolution of the simulated high-frequency absorption profile closely resembles the changes in BC profiles, since the aerosol absorption is primarily influenced by BC.

Lines 345-346, "Gas-aerosol exchange", is this gas-phase chemical reaction of SO2?

Yes, "Gas-aerosol exchange" refers to changes from the gas-phase oxidation of SO2. In the revised manuscript, we directly change it to "gas-phase production from $SO_2$ oxidation".

Lines 347-351, the analysis shown in Figure 9 is interesting. I think that small contribution of the below-cloud scavenging is also interesting. What scheme is used for the below-cloud scavenging in the model calculations?

EAMv2 follows the wet deposition scheme described in Wang et al. (2020). Below-cloud scavenging refers to the capture of interstitial (air-borne) aerosol by precipitation particles through Brownian diffusion or inertial impaction.

Figure 9, please describe the figure caption more carefully and clarify abbreviation (AQ chem H2SO4, SO4, etc.).

We have made the following changes to the figure caption for clarity:

"AQ chem (SO4)" refers to aqueous-phase chemical production through oxidation of SO2 by hydrogen peroxide and ozone. "AQ chem (H2SO4)" refers to aqueous-phase chemical production through cloud-water uptake of H2SO4. "in-cloud" wet deposition refers to the nucleation scavenging and "below-cloud" wet deposition is the washout (or impaction scavenging) by rain or snow.

Figure 12 and Figure S10, RMSE of POM in LR simulations (Figure S10b and S10d) are much greater than those in PRM simulations (Figure 12b and 12d). On the other hand, RMSE of BC and SO4 are similar levels in all simulations (Figures 12, 14, S10, S11). Could the authors explain this, if possible?

Thank you for pointing out the problem. This is due to a sampling error. We have now made sampling consistent for all cases. We have revised Figure S10 (shown as Figure R14 below). Revised RMSE values are within the expected range.

[Figure]

**Figure R14:** Scatter plots between simulated and observed monthly mean surface concentrations of (a, c) Black Carbon (BC) and (b, d) Primary Organic Matter (POM). Observations of the surface concentrations are from IMPROVE for the simulation year of 2016. Scatter plot statistics compare the Spearman's correlation (R), number of data points (n), RMSE, NMB values between (a, b) LR-PD, (c, d) LR-EHR-PD, and (e, f) LR-SE-PD simulation. RMSE and NMB are defined in Table 2. Solid lines indicate the 1:1 ratio, and the dashed lines indicate the 1:2 and 2:1 ratio. The values at the top of each column indicate the observed mean.

Lines 452-471, is one-year integration sufficient to extract the difference in aerosol radiative forcings, specifically aerosol-cloud interactions, between the revised and default emission treatments? The smaller difference over NA (compared to that over the ocean) does not guarantee a validity of the analysis? Could the authors answer this, if possible?

In our simulations, we apply nudging to constrain the large-scale circulation so that the aerosol forcing signals can be identified using short one-year simulations (Zhang, et al. 2022). On the other hand, over the low-latitude regions (tropics and sub-tropics) nudging cannot constrain

the circulation as efficiently as over the mid-latitude and polar regions (Sun, et al. 2019). This is one major reason why we see some noisy patterns that show large neighboring positive and negative values (e.g., 15°N-30°N, 140°W-170°W). Furthermore, the atmosphere is more pristine over the ocean compared to that over the land, so it is more sensitive to anthropogenic aerosol perturbations. Therefore, a larger difference over some ocean areas is possible.

We believe the nudged simulations can provide a reasonable estimate of the regional mean aerosol forcing for a relatively large region as chosen in this study and the overall forcing pattern. However, for a more accurate estimate of the aerosol forcing values in individual grid boxes, much longer simulations or large ensembles of short simulations are needed.

Lines 501-502, I cannot not understand this. The authors state here the importance of the spatial heterogeneity for ACI. On the other hand, they state that the overall impact of the revised emission treatment is small for the anthropogenic aerosol forcing estimates (Lines 470-471).

Figure R15 illustrates the difference in TOA forcing estimates between the revised and default emission treatment from LR and RRM simulations. As described in the response to the comment above, we see some large neighboring positive and negative differences over the subtropical and tropical ocean, which are more likely noises rather than signals. On the other hand, both LR and RRM simulations show some structural differences over the land and ocean areas (the signal will be more obvious if we smooth the data). Therefore, even though the mean differences are small, we may still find large regional/local impact. For clarity, we revised the paragraph in line 470 as below:

"For the annually-averaged regional mean anthropogenic aerosol forcing over NA, the relative difference is only about 3-5% between the two treatments. On the other hand, both LR and RRM simulations show structural regional differences over the land and ocean areas. Therefore, even though the mean differences in the entire NA region are small, the regional/local impact can still be large. Over the subtropical and tropical ocean (mostly away from anthropogenic aerosol emission sources), there are some large neighboring positive and negative differences, which are mostly caused by perturbations (noises) in the cloud fields. This is mainly due to the fact that nudging cannot constrain the large-scale circulation in those areas as efficiently as over the mid-latitude and polar regions (Sun, et al. 2019)."

[Figure]

**Figure R15:** Spatial distribution of anthropogenic aerosol radiative forcing differences (net: a,d; shortwave: b,e; and longwave: c,f) at the top of atmosphere (TOA) between simulations with revised and default treatment from (a-c) LR (ne30pg2) and (d-f) RRM simulations. "SW" and "LW" subscripts indicate shortwave and longwave forcing. Area-weighted regional mean differences are indicated at the upper-right corner of each panel.

**Technical comments:**

Lines 94-95, typo? "Fig. 1b", is this Fig. 1a?

Yes, changed to 1a.

Line 243, SO4 emissions→SO2 emission or sulfur emissions

EAM prescribes 2.5% of sulfur emissions as primary sulfate aerosol emissions following the AeroCom protocol (Dentener, et al. 2006). We added the following statement in section 2.2 for clarity:

"All prescribed emissions of BC and POM are considered as primary carbon mode aerosol particles (Liu et al., 2016). EAM assumes 2.5% of sulfur emissions as primary sulfate aerosol emissions following the AeroCom protocol (Dentener et al., 2006). Sulfate ($SO_4$) aerosol

particles are prescribed as either Aitken or accumulation mode based on the emission sectors or types."

Table 2 displays error statistics of prescribed primary sulfate aerosol emissions. "$SO_4$ emission" is modified to "$SO_4$ aerosol emission" for clarity.

Line 244, sulfate emission→sulfur emission or SO2 emission

Modified to "sulfate aerosol emission" for clarity.

Line 258, spatial distribution of "annual mean" surface concentration resulting ...

Added "annual mean".

Line 345, sulfate emissions, is this SO2 emissions or sulfur emissions?

Modified to "sulfate aerosol emission" for clarity.

Line 390, "Prescribed sulfate aerosol emissions", is this sulfate produced by SO2?

E3SM uses 2.5% of sulfur ($SO_2$) emissions as primary sulfate aerosol emissions.

Line 414, typo? "from 0.44 to 0.59 for BC and 0.43 to 0.51 POM"→"from 0.43 to 0.59 for BC and 0.44 to 0.51 POM"

Updated with revised values.

Lines 458-463, typo, several Fig. S13→Fig, S12

Fixed typo.

Line 466, typo, Fig. S14a, Figure S14d→Fig. S13a, Fig. S13d

Fixed typo.

**Works Cited**

Dentener, Franciscus, S Kinne, T Bond, O Boucher, J Cofala, S Generoso, P Ginoux, et al. 2006. "Emissions of primary aerosol and precursor gases in the years 2000 and 1750 prescribed data-sets for AeroCom." *Atmospheric Chemistry and Physics* 4321-4344.

Jones, Philip W. 1999. "First-and second-order conservative remapping schemes for grids in spherical coordinates." *Monthly Weather Review* 2204-2210.

Li, Jianfeng, Kai Zhang, Taufiq Hassan, Shixuan Zhang, Po-Lun Ma, Balwinder Singh, Qiyang Yan, and Huilin Huang. 2023. "Assessing the Sensitivity of Aerosol Mass Budget and Effective Radiative Forcing to Horizontal Grid Spacing in E3SMv1 Using A Regional Refinement Approach." *Geoscientific Model Development Discussions* 1-43.

McDuffie, Erin E, Steven J Smith, Patrick O'Rourke, Kushal Tibrewal, Ch Venkataraman, ra, Eloise A Marais, et al. 2020. "A global anthropogenic emission inventory of atmospheric pollutants from sector-and fuel-specific sources (1970--2017): an application of the Community Emissions Data System (CEDS)." *Earth System Science Data* 3413-3442.

Zhang, Kai, Wentao Zhang, Hui Wan, Philip J Rasch, Steven J Ghan, Richard C Easter, Xiangjun Shi, et al. 2022. "Effective radiative forcing of anthropogenic aerosols in E3SM version 1: historical changes, causality, decomposition, and parameterization sensitivities." *Atmospheric Chemistry and Physics* 9129-9160.

Sun, Jian, Kai Zhang, Hui Wan, Po-Lun Ma, Qi Tang, and Shixuan Zhang. 2019. "Impact of nudging strategy on the climate representativeness and hindcast skill of constrained EAMv1 simulations." *Journal of Advances in Modeling Earth Systems* 3911-3933.

Wang, Hailong, Richard C Easter, Rudong Zhang, Po-Lun Ma, Balwinder Singh, Kai Zhang, Dilip Ganguly, et al. 2020. "Aerosols in the E3SM Version 1: New developments and their impacts on radiative forcing." *Journal of Advances in Modeling Earth Systems.*